# The *Salmonella* pathogenicity island 1 injectisome reprograms host cell translation to evade the inflammatory response

George Wood [1], Rebecca Johnson[1], Jessica Powell[1], Owain J. Bryant [1], Filip Lastovka [1], Matthew P. Brember[1], Panagiotis Tourlomousis [2], John P. Carr [3], Clare E. Bryant [2] ✉ & Betty Y. W. Chung [1] ✉

During bacterial infection both the host cell and its invader must rapidly divert resources to synthesize specific proteins. For the host, these factors may be needed for innate immune responses, including programmed cell death, and in the bacteria newly synthesized proteins may include survival factors that counteract host defences. *Salmonella* is an important bacterial pathogen that invades and multiplies within host cells. It is well established that epithelial cell invasion is dependent upon the *Salmonella* pathogenicity island 1 (SPI-1) type III injectisome, a biological needle that penetrates host cells and injects effectors that promote bacterial internalization. However, the importance of the SPI-1 injectisome in infection of professional phagocytes such as macrophages, the predominant host cell type supporting systemic infection, is less clear. Through time-resolved parallel transcriptomic and translatomic studies of macrophage infection, we reveal SPI-1 injectisome-dependent infection of macrophages triggers rapid translation of transcription factors, including Early Growth Response 1 (EGR1). Despite EGR1's short half-life, its swift synthesis, driven by untranslated regions of its mRNA, is sufficient to inhibit the transcription of pro-inflammatory genes; this restrains inflammation and macrophage death which would otherwise abort systemic infection. This demonstrates the importance of translational activation in host–pathogen dynamics during bacterial infection.

The gene expression profiles of both host[1–5] and pathogen[6] are altered dramatically when they interact. This response is shaped by a co-evolutionary arms race where the host seeks to detect and counter the invading pathogen[3,5], whilst the pathogen aims to evade this response[6] and modify the host environment to better suit its survival and replication[1,2]. Changes in gene expression during infection are thus the net result of these competing goals, the balance of which will ultimately determine the infection outcome.

Many members of the Gram-negative bacterial genus *Salmonella* are facultative intracellular pathogens that infect a diverse spectrum of hosts. *Salmonella* species cause a range of diseases in humans, from typhoid fever to gastroenteritis. *Salmonella*'s ability to establish an intracellular infection is key to its pathogenesis[7]. During infection, the bacterium invades host cells, including the epithelial cells lining the intestinal tract and immune cells such as macrophages. Once internalized, *Salmonella* resides in a specialized membrane-bound compartment, the *Salmonella*-containing vacuole (SCV). The SCV provides a protective niche for the pathogen while giving it access to host cell nutrients to support its replication[8,9].

[1]Department of Pathology, University of Cambridge, Cambridge, UK. [2]Department of Medicine and Department of Veterinary Medicine, University of Cambridge, Cambridge, UK. [3]Department of Plant Sciences, University of Cambridge, Cambridge, UK. ✉e-mail: ceb27@cam.ac.uk; bcy23@cam.ac.uk

The intracellular lifestyle of pathogenic *Salmonella* is supported by two type III secretion systems (T3SS) with different substrate specificities. The SPI-1 T3SS, also called the SPI-1 injectisome (hereafter, injectisome) is a multi-protein complex that spans the bacterial inner and outer membranes and cell wall, and transports proteins into target cells along a homomeric needle-like structure that is inserted into the host cell membrane by the bacterial tip and translocon components SipB, SipC and SipD[10–12]. PrgJ forms the inner rod that connects the injectisome basal body embedded in the *Salmonella* envelope with the needle structure and thus is integral in connecting the channel down which effectors are transported[10,13]. In epithelial cells, the injectisome is important in aiding host cell invasion via the trigger mechanism[14–16]. The role of the injectisome in establishing infection in professional phagocytes, such as macrophages, is less well understood, though is markedly enhanced by the injectisome[14–18]. The SPI-2 T3SS is expressed once the bacterium is internalized and supports the intracellular lifecycle of the pathogen[8,19].

To allow transport of effector proteins via the injectisome into the host cell, the *Salmonella* translocon subunits SipB and SipC are also secreted via the injectisome and inserted into the host cell plasma membrane to form a transient pore[11,12,20,21]. This insertion leads to transient loss of membrane integrity, therefore triggering a plethora of stress responses, including collapse of ion gradients with $Ca^{2+}$ influx, and $Cl^-$ and $K^+$ efflux. Injectisome-dependent membrane damage is demonstrated by the haemolysin activity of the injectisome on red blood cells, which lyse upon SipBC insertion into their membranes[12]. While it is known that loss of membrane integrity, and the associated disruption of ion gradients, triggers inflammatory response pathways and cell death[20,22,23], the importance of this damage in eliciting host responses to *Salmonella* remains unclear.

In systemic infection, macrophages are the predominant host cell type and *Salmonella* survival in macrophages has been reported to be critical for virulence[7,24]. This is somewhat paradoxical given the importance of macrophages for the detection and elimination of pathogens[25]. Indeed, macrophages possess many receptors that detect pathogen associated molecular patterns (PAMPs), which are crucial in controlling infection. This includes toll-like receptors (TLRs) such as TLR4, which detects bacterial lipopolysaccharide (LPS) on the *Salmonella* cell surface[26], as well as intracellular immune receptors that can activate the inflammasome and lead to inflammatory cell death of macrophages[27–29]. In tissue culture, the majority of macrophages die within the first few hours of infection[30–32], and this rapid cytotoxicity is dependent on injectisome activity[30,32]. The balance between macrophage survival and death will influence the outcome of infection[25].

Although much research has been carried out on the transcriptional response of host cells to *Salmonella* infection[2–4], the analysis of gene expression is incomplete without also exploring regulation of protein synthesis, i.e., translational regulation. This is of particular importance given the critical nature of events occurring very early in infection. Translational regulation has the potential to allow for rapid responses, through modulating translation of pre-existing mRNAs and/or by enhancing translation of the newly transcribed mRNAs. Indeed, previous studies suggest that inflammatory genes, such as *Tnf*, can undergo translational regulation in macrophages following stimulation of TLR4 with purified LPS[33,34].

Here we paired ribosome profiling with RNA-Seq during infection of macrophages by wild type (WT) or SPI-1 injectisome mutant *Salmonella* to understand the regulation at the transcriptional and translational levels of the dynamics of gene expression driven by injectisome-dependent infection. We discovered that the translational response precedes the transcriptional response, and that it enhances production of transcription factors. One such transcription factor was Early Growth Response 1 (EGR1). Although there was rapid injectisome-dependent transcription of *Egr1*, there was an even greater induction of its translation, enabling the rapid and robust production of the EGR1 transcription factor early in infection. We further demonstrated that while EGR1 protein turnover is rapid, it acts as a longer-term transcriptional suppressor of inflammatory genes triggered by *Salmonella* infection. We conclude that this creates an early window of opportunity for *Salmonella* to circumvent innate immunity and to successfully establish infection.

## Results

### *Salmonella* infection triggers rapid host cell responses

*Salmonella* entry into immortalized murine bone marrow derived macrophages (iBMDM, hereafter macrophages unless otherwise stated) is markedly enhanced by the injectisome[14–17]. *Salmonella* cells associated with the host cell membrane, that is bacteria actively invading the host, were seen as early as 5 min post exposure, with *Salmonella* rapidly internalized within 15 min (Fig. 1A, B). There was little further macrophage infection after 15 min, indicating a potential change in susceptibility (Fig. S1A). Injectisome-independent internalization was assayed through infection with mutant *Salmonella* unable to assemble the injectisome needle, generated by knockout of the SPI-1 inner-rod protein PrgJ[35] (Δ*prgJ*) and referred to as the injectisome mutant hereafter. Invasion by injectisome mutant *Salmonella* was also observable within this timeframe, though to a lesser degree than by WT bacteria (Fig. S1B).

As previously described[30–32], infection of macrophages with WT *Salmonella* led to rapid cell death within the first 60 min of infection. However, ~25% of macrophages survived despite the presence of viable intracellular bacteria (Fig. 1C, D, S1A). In contrast, and similarly to previous studies[36], infection with injectisome mutant *Salmonella* did not induce any increase in macrophage cell death (Fig. 1D).

### Injectisome-dependent infection leads to both transcriptional and translational induction of Early Growth Response 1 (EGR1)

As a significant proportion of macrophages survive the lethal effect of the injectisome (Fig. 1A–D), we hypothesized that survival of infected cells beyond 60 min post infection with WT *Salmonella* may be a consequence of rapid gene expression responses occurring within the first hour. This may be mediated by de novo transcription of mRNAs, which has been the focus for most previous studies[2–4]. However, we suspected the involvement of translational responses, which remain much less well understood, where the rate of protein synthesis (translational efficiency) for certain mRNAs may be specifically upregulated to increase protein abundance more rapidly than could occur via transcription alone. Such an acceleration of protein synthesis could be of critical importance given the short survival timeframe of most infected cells. To investigate rapid transcriptional and translational changes in gene expression, parallel global transcriptomic (RNA-Seq) and translatomic (ribosome profiling; Ribo-Seq hereafter) analyses were performed on macrophages at 60 min post infection with either WT *Salmonella* or the injectisome mutant (Fig. 1E, S1C). Ribo-Seq is a highly sensitive method that reveals the global translatome at the time of harvest[37]. The technique determines the position of ribosomes by exploiting the protection from nuclease digestion of a discrete fragment of mRNA (~30 nucleotides) conferred by elongating ribosomes. Deep sequencing of these ribosome-protected fragments (RPFs) generates a high-resolution view of the location and abundance of translating ribosomes on different mRNA species, reflecting the amount of synthesis of specific proteins. In addition, while RNA-Seq enables quantification of total mRNA abundance, parallel RNA-Seq and Ribo-Seq enable quantification of translation efficiency, a measurement of how well each mRNA is being translated, as distinct from total protein synthesized (Fig. 1F).

The resolution of our Ribo-Seq data is high, as is evident from the metagene analysis constructed with the R package riboSeqR[38]. The metagene translatome (Fig. S1C) shows the weighted average number of reads around the annotated coding start and stop sites of all

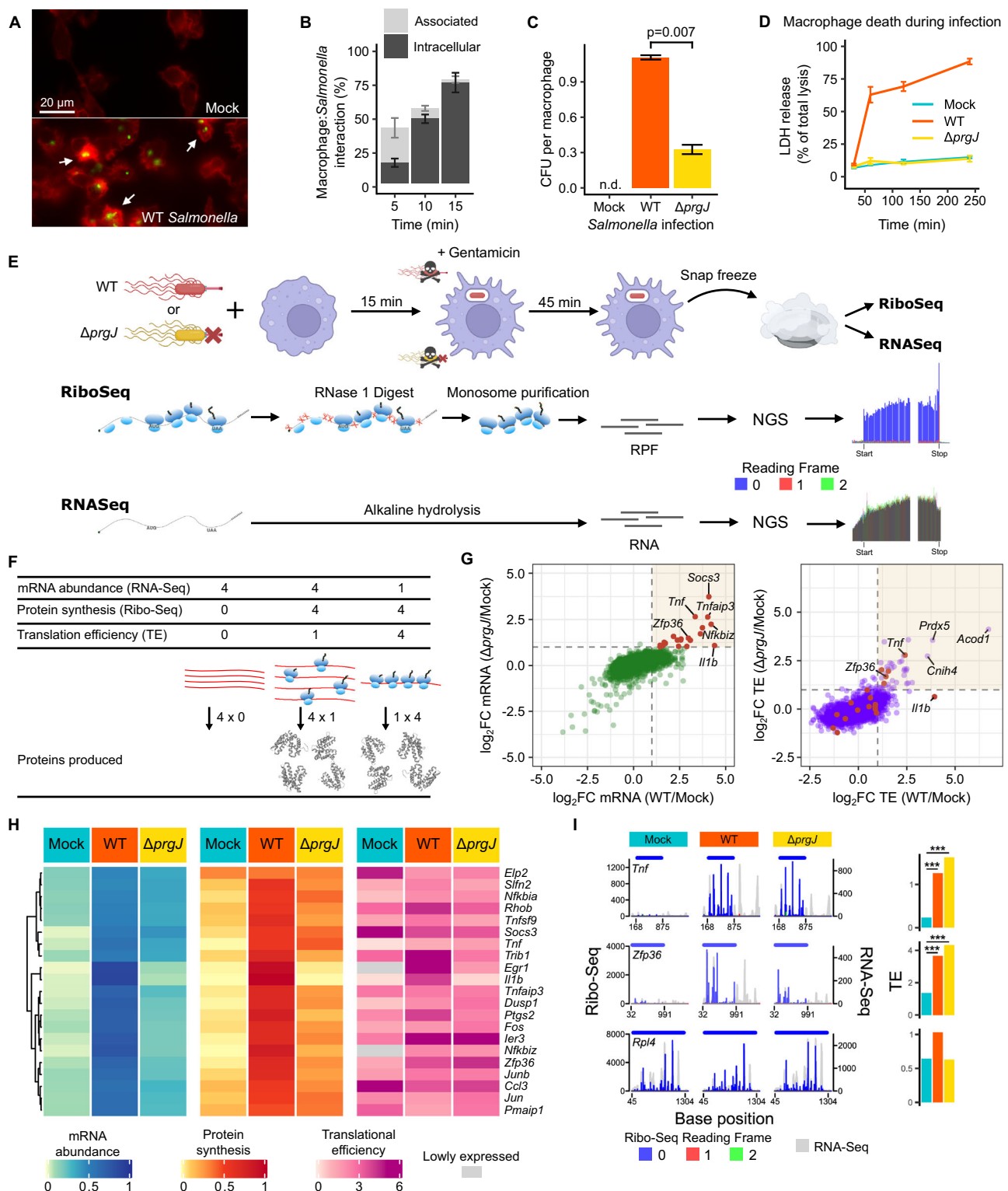

translated mRNAs. Ribo-Seq reads are exclusive to coding regions and demonstrate the accurate capture of the triplet periodicity of the elongating ribosome movement as the 5′ end of almost all Ribo-Seq reads map within the same reading frame relative to the first codon position of the coding sequence. This high resolution is further evidenced by our ability to directly visualize translation of single genes at remarkable accuracy, as well as non-canonical translation events. For example, the data captures translation of *Atf4*, which is modulated by translation of two small upstream open reading frames (uORFs) embedded within the 5′UTR of the *Atf4* transcript[39] (Fig. S1E). The high-

resolution nature of our data therefore enables accurate quantification of protein synthesis (i.e., total Ribo-Seq) and translational efficiency when combined with parallel RNA-Seq (Fig. 1F).

Initial analysis of macrophages infected by both WT and injectisome mutant *Salmonella* identified many genes known to be upregulated upon exposure to bacterial PAMPs[33,34,40] (Fig. 1G–I, S1D). Many of these genes, such as *Tnf* and *Zfp36*, were upregulated not only in transcript abundance but also at the translational level. The transcription of these genes was induced in response to either WT or injectisome-mutant *Salmonella* and potent translational

**Fig. 1 | *Salmonella* infection rapidly alters macrophage translation. A** Mock-infected and WT *Salmonella*-infected macrophages 5 min post infection. Actin (red); *Salmonella* (green). Arrows indicate membrane-associated bacteria. **B** Percentage of macrophages interacting with WT *Salmonella* during the first 15 min, as determined by immunofluorescence microscopy and stratified by membrane-associated or intracellular bacteria. Three biological replicates. **C** *Salmonella* colony forming units (CFU) recovered 75 min post mock, WT and SPI-1 deficient (Δ*prgJ*) *Salmonella* infection per macrophage. As expected, no CFUs were detected in mock infection (n.d.). Significance determined by two-sided Student's *t*-test; two biological replicates; three technical triplicates each. **D** Cytotoxicity of WT or Δ*prgJ Salmonella*, or mock infection of macrophages as determined by lactate dehydrogenase (LDH) release into culture supernatant. Four technical replicates, representative of 3 independent experiments. **B**–**D** Error bars show mean ± SEM. **E** Experimental outline detailing the preparation of ribosome profiling and RNA-Seq libraries from infected macrophages. Details are provided in the materials and methods (RPF: ribosome-protected RNA fragments, NGS: next-generation sequencing). **F** Diagram illustrating the relationship between mRNA abundance (RNA-Seq), protein synthesis (Ribo-Seq), and translational efficiency (TE).

**E**, **F** Created in BioRender. Chung, B. (2025) https://BioRender.com/sjej7aq. **G** Changes in mRNA abundance (left) and TE (right) on infection with WT or Δ*prgJ Salmonella* compared to mock infection. Genes upregulated transcriptionally in both (log₂FC > 1) are shown in red. **H** Normalized relative mRNA abundance, protein synthesis, and TE of transcriptionally upregulated mRNAs from (**G**). Hierarchical clustering was used to order genes by mRNA abundance across all conditions (left). Genes with low read counts (i.e., normalized RNA-Seq and Ribo-Seq counts <5; shown in grey) are too poorly expressed to reliably calculate TE values. **I** Normalized Ribo-Seq read count and RNA-Seq coverage of *Tnf* and *Zfp36*, which are known to have increased TE on exposure to bacterial PAMPs. Ribo-Seq reads are represented by their P site position, colored by their reading frame relative to the start of the coding sequence, represented by the bar above each plot with the start and stop positions indicated on the x-axis. *Rpl4* is presented as a control gene expressed in all samples. *P* values were estimated using the two-tailed Bayesian framework from Xtail, comparing mRNA and RPF log₂FC and log₂ratios across two pipelines and reporting the more conservative result with Benjamini-Hochberg correction for multiple comparisons[87]. All adjusted *P* < 10⁻¹².

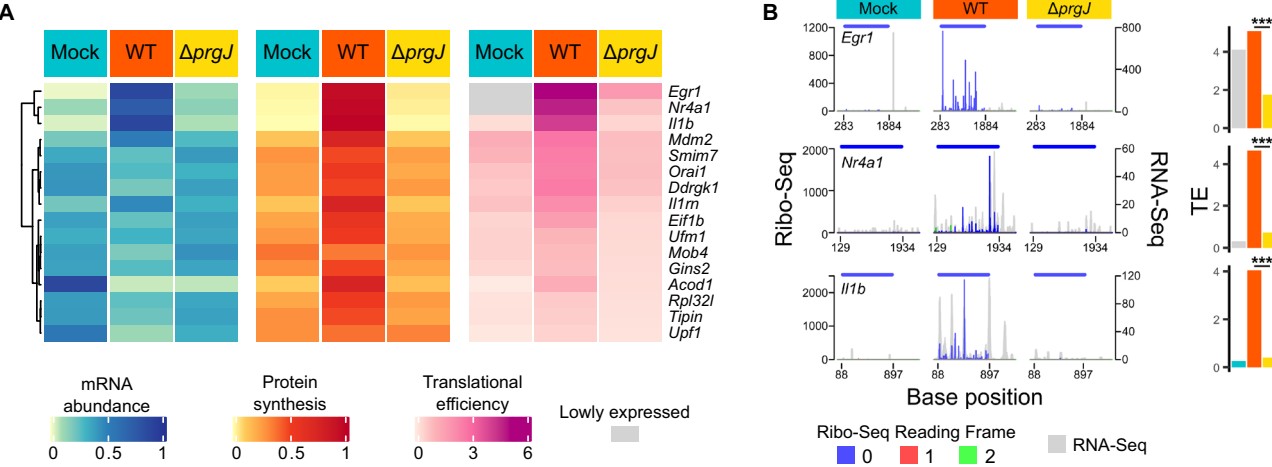

**Fig. 2 | *Salmonella* SPI-1 dependent infection triggers a distinct host translational program. A** Normalized relative mRNA abundance, protein synthesis, and TE of mRNAs with log₂FC in TE of WT over Δ*prgJ* infected macrophages greater than 1.5 and at least 50 normalized Ribo-Seq counts in WT infection. Transcripts are ordered by hierarchical clustering of TE across all conditions (left). **B** Normalized Ribo-Seq read count and RNA-Seq transcript coverage of *Egr1*, *Il1b* and *Nr4a1* as in

Fig. 1G. Genes with low read counts (i.e., those with normalized RNA-Seq and Ribo-Seq counts <5; shown as grey bars in **B**) are too poorly expressed to permit TE values to be reliably calculated. *P* values were estimated using the two-tailed Bayesian framework from Xtail, comparing mRNA and RPF log₂FC and log₂ratios across two pipelines and reporting the more conservative result with Benjamini-Hochberg correction for multiple comparison[87]. All adjusted *P* < 10⁻¹⁴.

upregulation was also observed (Fig. 1G–I). Enhanced translational efficiency results in greater protein synthesis than can be explained by an increase in de novo mRNA synthesis alone (Fig. 1F), and has been previously reported in macrophages exposed to purified bacterial LPS[33,34].

Following this, transcripts that were subject to specific injectisome-dependent upregulation were investigated, that is they are upregulated in WT infection compared to the injectisome-mutant *Salmonella* infection (Fig. S1F). The key inflammasome component *Nlrp3* was among the selectively induced genes (Fig. S1G). The human *NLRP3* transcript has recently been described to encode a uORF[41], and our data indicate that this is also true for the mouse *Nlrp3* mRNA (Fig. S1H). The NLRP3 protein is highly activated in *Salmonella* infection, although the uORF-mediated translational regulation of *Nlrp3* is currently unclear[35,42,43]. Our data revealed that while protein synthesis of *Nlrp3* is enhanced in an injectisome-dependent manner, translation of the uORF is activated to prevent the deleterious consequences of NLRP3 protein overproduction[44]. To further identify injectisome-dependent translationally regulated genes, transcripts where the log₂ fold-change (log₂FC) in translational efficiency was greater than 1.5 in WT *Salmonella*-infected macrophages over macrophages infected by the injectisome mutant were selected (Fig. 2A). Amongst the most

translationally upregulated mRNAs were those encoding EGR1, NR4A1, and Pro-interleukin-1β (pro-IL-1β) (Fig. 2).

Interleukin-1 beta (IL-1β) is processed from its precursor pro-IL-1β into its mature, active form by caspases, proteases that are themselves activated by the inflammasome[29,45]. This precursor, encoded by *Il1b*, had a much greater translation efficiency in macrophages infected by WT *Salmonella* than by the injectisome mutant (Fig. 2). The injectisome is known to transport bacterial effectors into the host cytosol that activate the inflammasome, including components of the translocon such as SipB and SipC[46,47], consistently with our cytotoxicity assay (Fig. 1D), leading to IL-1β production[28,48]. These data suggest that IL-1β precursor production is supported not only by increased mRNA transcription, but also post-transcriptionally by specific translational upregulation of its mRNA. Previous reports describe the need for two signals to produce IL-1β: one to induce transcription of the precursor and another to activate the inflammasome[46,49,50]. Importantly, however, our data additionally reveal a strong role of translational upregulation to rapidly facilitate overall IL-1β precursor protein production. Therefore, the injectisome makes an important contribution to macrophage pro-IL-1β production within the first 60 min, in addition to delivering the factors that stimulate pro-IL-1β cleavage and lead to IL-1β production.

Both *Egr1* and *Nr4a1* are known immediate early genes that are rapidly transcribed in many cell types within minutes in response to a range of cellular stresses[51–53]. Here, we show that while transcripts for both genes are almost absent in uninfected macrophages, rapid expression of these genes is driven by specific transcriptional upregulation alongside a surge of specific translational activation following injectisome-dependent infection (Fig. 2). NR4A1, also known as NUR77, is involved in macrophage responses to proinflammatory stimuli. NR4A1 limits inflammation in models of sepsis and colitis, likely through antagonism of the NF-κB pathway[54,55]. More recently, however, it has been reported to increase expression of proinflammatory cytokines in mice infected with *Klebsiella pneumoniae*[56]. The role of NR4A1 is therefore certainly immunomodulatory but likely differs by cell type and context.

Of these transcripts chosen for detailed study, the most translationally induced is the mRNA encoding EGR1 (Fig. 2). EGR1 is a zinc-finger family transcription factor that binds GC-rich consensus sequences in gene promoters and enhancers, and it can either activate or suppress transcription. Targets of EGR1 span diverse biological processes including immune responses, cell growth and differentiation, and cell death[57–62]. Due to its highly injectisome-dependent translational upregulation, the biological function of EGR1 in *Salmonella*-macrophage infection was further characterized in this study.

## EGR1 protein accumulation during the macrophage–*Salmonella* interaction is highly injectisome-dependent

Transcriptional upregulation of *Egr1* has previously been shown to be largely dependent on bacterial secretion systems in other infection contexts, including those where the bacteria remain extracellular[2,63–65]. However, EGR1 has not been studied in the context of *Salmonella* infection of macrophages, nor has post-transcriptional regulation of *Egr1* expression been studied. We were particularly interested in EGR1 given its association with cell death[57] and its importance in macrophage development[52,60].

At 60 min post infection, there was a clear increase in both transcription and translation of *Egr1* in WT-infected cells (Figs. 2 and 3A). The increase in *Egr1* translation cannot be explained by the greater transcript abundance alone, but rather there was also an increase in *Egr1* mRNA translation efficiency (i.e., increase in ribosome occupancy per mRNA molecule). The translational efficiency of *Egr1* mRNA is three times higher in WT-infected cells compared to cells infected with the injectisome mutant. In contrast, the *Egr1* expression in cells infected with the injectisome mutant were not much greater at 60 min post infection than in mock-infected cells. Reverse transcription-coupled quantitative PCR (RT-qPCR) for *Egr1* mRNA and immunoblot for EGR1 protein accumulation showed that, during the infection time course, the upregulation was rapid but transient, and that the half-lives of both *Egr1* mRNA and its protein product are short. While *Egr1* mRNA abundance peaked at 60 min and returned to baseline levels by 120 min (Fig. 3B), the increase in protein abundance measured by immunoblotting was, as expected, offset from the increase in mRNA, peaking at 120 min post infection and returning to undetectable levels by 240 min (Fig. 3C, D). The absence of EGR1 protein 240 min post infection indicates a short half-life of EGR1 protein and that its biological effect is likely rapid. In contrast, infection with the injectisome mutant led to a slight increase in *Egr1* mRNA abundance at 60 min, followed by a detectable increase in EGR1 protein by 120 min, with both mRNA and protein at considerably lower levels than in WT *Salmonella*-infected cells (Fig. 3A–D).

## EGR1 restrains the macrophage inflammatory responses to *Salmonella* infection

While EGR1 has numerous reported functions in cell differentiation, death, and growth[57–62], its role in infection is poorly understood. Therefore, we generated EGR1 knockout (EGR1^KO) macrophages using

CRISPR-Cas9 and confirmed the absence of EGR1 protein 120 min after infection with WT *Salmonella* (Fig. 3E). Mutation of the *Egr1* coding sequence in EGR1^KO macrophages was confirmed through genomic DNA sequencing (Fig. S2A). Loss of the ability to produce EGR1 resulted in greater cell death at baseline, and this was further increased when infected by WT *Salmonella*, particularly between 30 min and 120 min post infection (Fig. 3F). This is despite equivalent susceptibility to infection (Fig. S2B). This suggests EGR1 may play a role in limiting injectisome-induced inflammatory macrophage death. Following this, the role of EGR1 in the inflammatory response was further assessed by measuring the level of IL-1β produced by EGR1^KO and EGR1^WT macrophages during infection (Fig. 3G). This revealed significantly greater upregulation in EGR1^KO macrophages infected with WT *Salmonella*, confirming that EGR1 has a suppressive role in regulating the inflammatory response. The greater levels of cell death in EGR1^KO macrophages may mean that the IL-1β production is underestimated in these cells and therefore the difference is likely even greater.

As EGR1 is known to act as a DNA-binding transcription factor, we hypothesized that EGR1 suppresses inflammation through transcriptional regulation and so time-resolved transcriptomic analysis of WT *Salmonella*-infected EGR1^KO or EGR1^WT macrophages from 15 to 240 min post infection was performed. Principal component analysis of gene transcript levels showed a clear separation of samples by cell line and time post infection. Principal component (PC) 1 largely reflects variation between the EGR1^KO and EGR1^WT cell lines, whereas PC2 separates the infected samples at 120 min and 240 min from the infected samples at 15 to 60 min and mock inoculation treatments 15–240 min (Fig. S2C). We observed a significant increase in the abundance of transcripts associated with immune responses in the EGR1^KO macrophages compared to the EGR1^WT macrophages, particularly at 240 min post infection, confirming the suppression of transcription of inflammatory genes by EGR1 during *Salmonella* infection (Fig. 3H–I; –2). This includes *Il1b*, which showed greater upregulation of transcription in EGR1^KO macrophages infected with WT *Salmonella*. This is likely the cause of the greater IL-1β secretion during *Salmonella* infection in the absence of EGR1 (Fig. 3G). Together, this confirms EGR1 as an important transcriptional immunomodulator. In addition, gene ontology enrichment analysis also revealed significant upregulation of known pro cell-death genes in infection of the EGR1^KO macrophages, consistent with a role of EGR1 in limiting macrophage death (Fig. 3F; Supplementary Data 1–2).

## Transcriptional and translational dynamics of *Salmonella* during macrophage infection

As shown above, macrophage expression of *Egr1* is tightly temporally regulated following its *Salmonella* SPI-1-dependent induction but the impact of its upregulation continues beyond the presence of EGR1 protein. Given the clear importance of rapid but transient gene expression, we next sought to characterize the temporal dynamics of the transcriptional and translational regulation during *Salmonella* infection. We therefore performed time-resolved parallel RNA-Seq and Ribo-Seq of primary bone marrow-derived macrophages infected with WT *Salmonella* or the injectisome mutant (Fig. 4A). The use of primary macrophages (rather than the iBMDMs used in previous experiments) allows us to compare injectisome- versus PAMP-dependent responses in a system that more closely resembles in vivo infection. That said, primary macrophages behaved similarly to the iBMDMs with rapid host cell death in WT *Salmonella* infection (Fig. 4B). The primary macrophages also showed similar rates of infection by WT *Salmonella*. In injectisome mutant infection, a greater proportion of primary macrophages contained the injectisome-mutant compared to iBMDMs but still fewer than in WT *Salmonella* infection (Figs. S1B and S3A). This may reflect a difference in their susceptibility to injectisome-independent means of host cell entry. Indeed, primary monocyte-

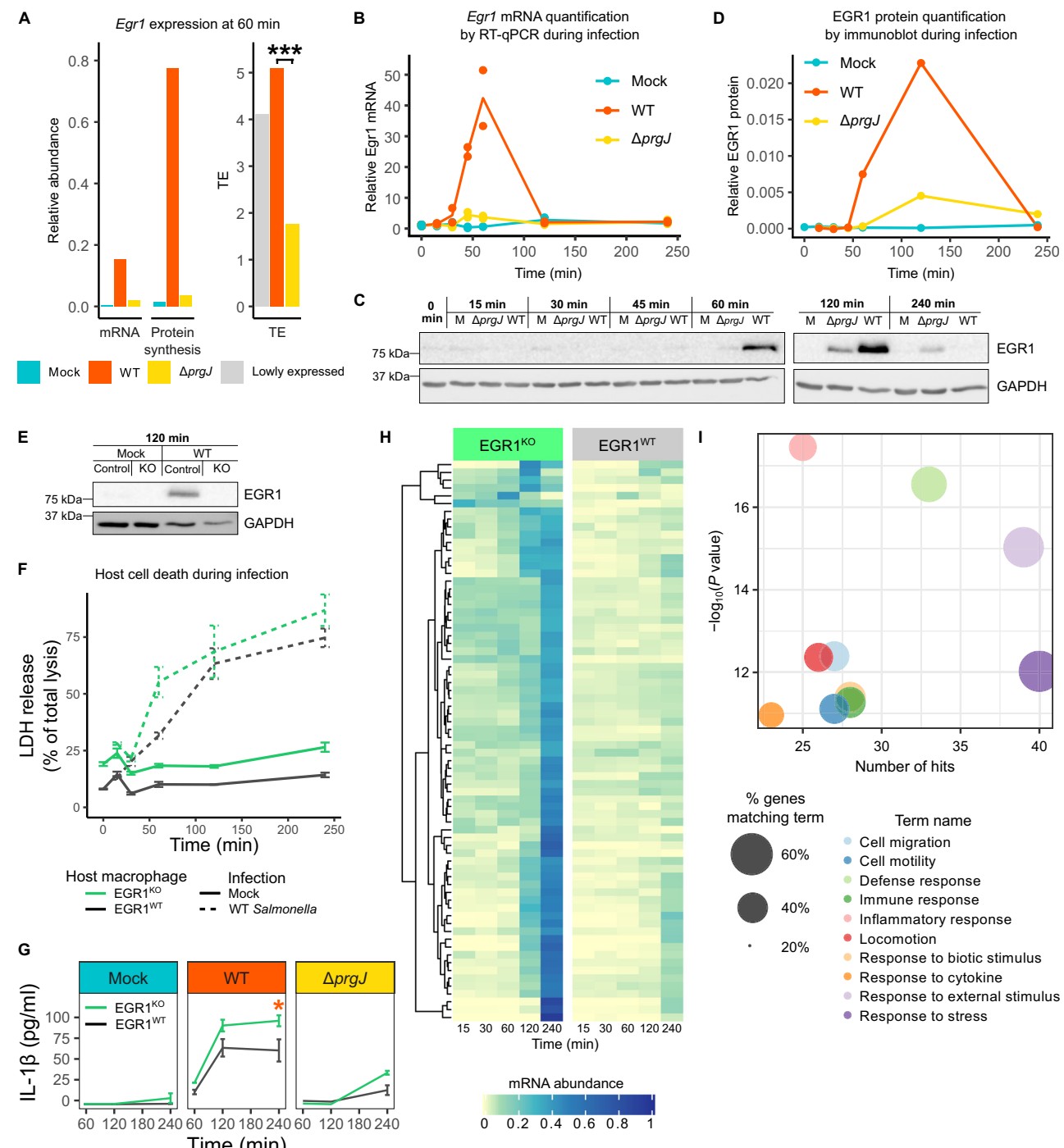

**Fig. 3 | *Egr1* is rapidly induced in *Salmonella* infection to restrict expression of immune response genes. A** mRNA abundance, protein synthesis, and translational efficiency (TE) of *Egr1* from Fig. 1K. In conditions with low *Egr1* expression TE cannot be reliably calculated (gray bar). *P* values were estimated using the two-tailed Bayesian framework from Xtail, comparing mRNA and RPF log₂FC and log₂ratios across two pipelines and reporting the more conservative result with Benjamini-Hochberg correction for multiple comparison[87]. $P = 1.02 \times 10^{-15}$ **B** *Egr1* transcript abundance during WT and SPI-1 deficient mutant (Δ*prgJ*) infection, normalized to the housekeeping gene *Supt16* and relative to their abundance in cells prior to infection; *n* = 2. **C** Immunoblot following EGR1 and GAPDH protein abundance across WT and Δ*prgJ* infection time course. An equal amount of cellular protein was loaded per lane. Samples derive from the same experiment and blots were processed in parallel. Representative of 2 independent experiments. **D** Quantification of EGR1 abundance from **C** normalized to GAPDH. **E** Immunoblot showing EGR1

expression in WT (EGR1^WT) and EGR1 knockout (EGR1^KO) macrophages infected with WT *Salmonella* at 120 min post infection. Representative of 2 independent experiments. **F** Cytotoxicity of WT or mock infection of EGR1^WT and EGR1^KO macrophages determined by lactate dehydrogenase (LDH) release into culture supernatant; *n* = 3. **G** IL-1β concentration in culture supernatant from infected EGR1^KO and EGR1^WT macrophages. 2 biological replicates except for 240 min WT *Salmonella* infection where there were 3 for EGR1^WT and 4 for EGR1^KO; significance determined using one-sided Student's *t*-test; *P* = 0.048; error bars show mean ± SEM. **H** Normalized mRNA abundance of transcripts upregulated (log₂FC > 2) in EGR1^KO compared to EGR1^WT macrophages at any time point in WT *Salmonella* infection. Genes are ordered by hierarchical clustering (left). **I** The 10 most significantly enriched gene ontology biological process terms in genes identified in H. *P* values were determined with Fisher's one-tailed test with Benjamini-Hochberg correction for multiple comparisons.

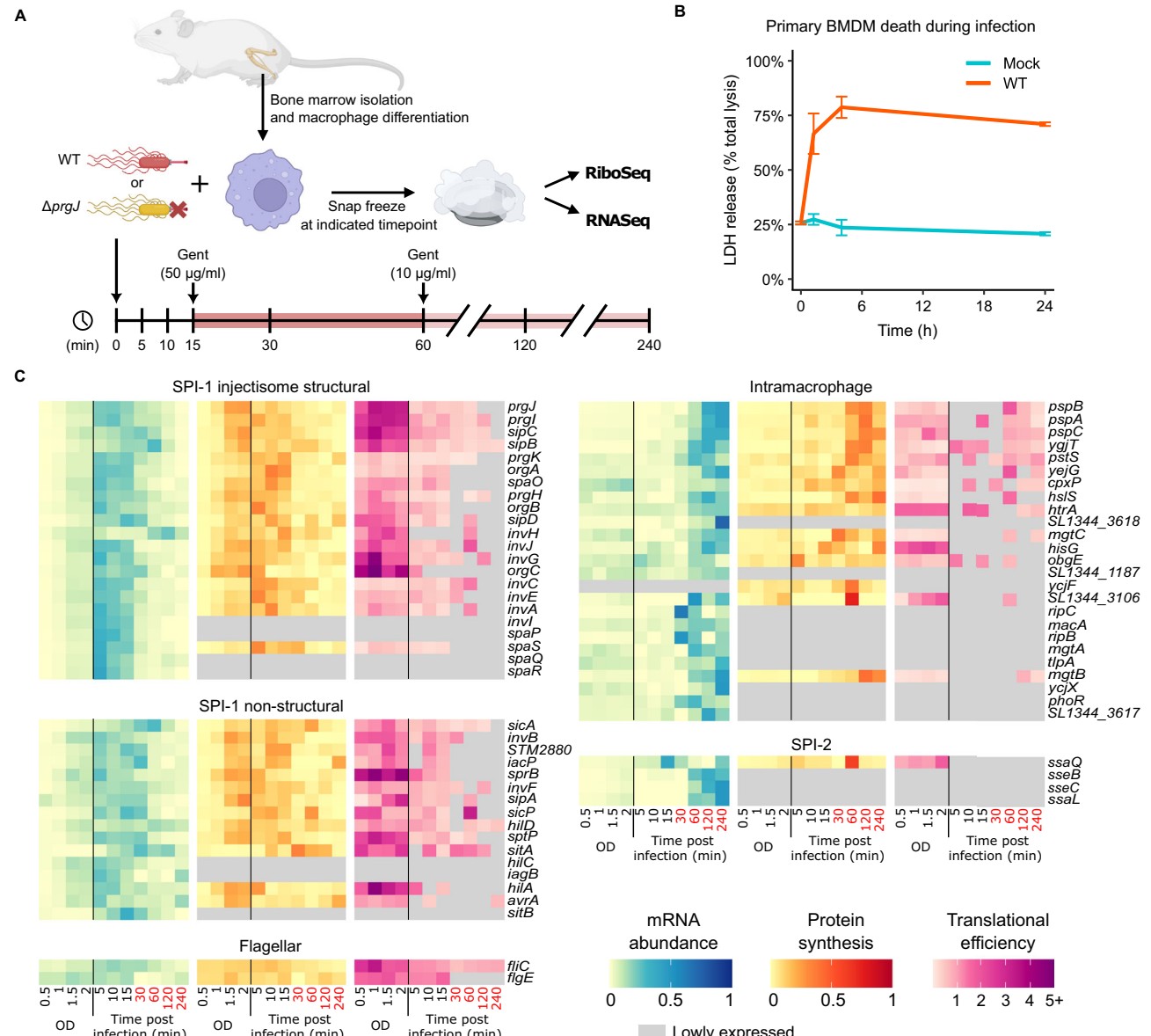

**Fig. 4 | Time-resolved Ribo-Seq reveals dynamic changes in bacterial translation.** Data from two biological replicates. **A** Outline of infection time course. Primary bone marrow-derived macrophages were generated from mice and infected with WT or Δ*prgJ* mutant *Salmonella*. Gentamicin was used to kill extracellular bacteria at 15 min, and its concentration was reduced after 1 h. Created in BioRender. Chung, B. (2025) https://BioRender.com/sjej7aq. **B** Cytotoxicity of WT *Salmonella* or mock infection of primary macrophages determined by lactate dehydrogenase (LDH) release into culture supernatant; 2 mice, technical duplicates. **C** Transient expression dynamics of select WT *Salmonella* genes at different *Salmonella* culture optical densities (OD) (Supplementary Data 6) and during the 4 h macrophage infection, at the level of mRNA abundance, protein synthesis, and translation efficiency.

derived macrophages have been shown to be more phagocytic than immortalized cell lines[66,67].

Due to the nature of the mRNA enrichment and the library preparation method employed, both host and bacterial translational and transcriptional dynamics were simultaneously captured over the course of the infection. Transcription and translation were also assessed in *Salmonella* grown in Luria Broth (or lysogeny broth, LB) at a range of culture densities (measured as optical density at 600 nm; OD). Capturing the gene expression of *Salmonella* at fine temporal resolution in this way is important because, as demonstrated above, the host response is tightly linked to the expression of bacterial virulence factors which themselves may undergo temporal regulation during infection. Indeed, it has been established that expression of SPI-1 in LB culture peaks in the late exponential phase, at OD 2, to which they were cultured in these infection assays[32]. These data also revealed that this increase in

expression of SPI-1 injectisome structural genes as the bacteria approach OD 2 is primarily through enhanced translation (Fig. 4C). Remarkably, we see significant transcriptional induction of SPI-1 genes upon contact with macrophages, particularly those encoding structural components such as PrgI, PrgJ and PrgK, as early as 5–15 min post infection, aligning with the timeframe of the initial *Salmonella*–macrophage interaction (Fig. 1A–C). This indicates that *Salmonella* responds to proximity to macrophages by upregulating expression of certain genes that enable invasion (Fig. 4C) and suggests that rapid assembly of injectisomes occurs on the bacterial cell surface during the first 15 min. Indeed, the rapid assembly of additional T3SSs upon contact with host cells has also been suggested to occur in *Yersinia enterocolitica* infection[68]. The transcriptional and translational response of non-structural SPI-1 genes that encode effector proteins, structural genes with a dual-role as effectors such as *sipB* and *sipC*, and other virulence factors also increased during

the first 15 min. In contrast, certain other SPI-1 genes, such as *avrA* and *iacP*, were maximally expressed between 2–4 h post infection (Fig. 4C).

Notably, during the early stages of infection (within the first 30 min), we also observed increases in the translation efficiency of a subset of intramacrophage genes, including *htrA*, *hisG* and *cpxP* (Fig. 4C). Intramacrophage genes play a crucial role in the survival and replication of the bacterium inside macrophages and in general maximal transcription of these genes occurs later. The functions of these genes include roles in magnesium and phosphate transport and the envelope stress response[69,70]. At the 30 min and 60 min time points, when *Salmonella* is intracellular and after SCV acidification[71], the transcript abundance of SPI-2 genes increases while the abundance of the majority of SPI-1 transcripts decreases (Fig. 4C). This agrees with the recognized switch of *Salmonella* secretion mediated by the transcriptional regulator SsrB. Concurrently, we observed decreases in expression of the flagella components, consistent with previously reported SsrB-mediated repression[69,72,73] (Fig. 4C).

## A global translational response precedes transcriptional responses in macrophages following injectisome penetration

Alongside this fine temporal profiling of *Salmonella* gene expression, analysis of parallel Ribo-Seq and RNA-Seq throughout the primary macrophage infection time course provided a global perspective of the dynamics of host gene expression at multiple levels, notably changes in RNA abundance and translation efficiency. Within 5 min post infection, we can readily visualize that both WT and injectisome mutant *Salmonella* induced rapid changes in the translation efficiency of many host genes compared to mock infection, with more genes that are translationally induced in an injectisome-dependent manner (Fig. 5A, S4B). A modest anticorrelation was also observed between log$_2$FC in TE and mRNA abundance, likely reflecting the temporal lag between transcriptional and translational responses, as newly synthesized transcripts require nuclear export and ribosome loading before measurable changes in TE can occur. This highlights the importance of stringent fold-change cutoffs when defining differentially regulated genes. Furthermore, comparison of WT *Salmonella* and injectisome mutant infections revealed that the injectisome-specific translational regulation precedes the transcriptional regulation, with relatively little change in mRNA abundances within the first 120 min (Fig. 5B, C). Many components of the classical inflammasome that are activated by *Salmonella*[42], including *Nlrp3*, *Casp1* and *Gsdmd*, are among those transcriptionally upregulated after 120 min of infection with either strain of *Salmonella* (Fig. S4C). This is after the initial wave of cell death following WT infection (Fig. 1C) that is attributed to inflammasome activation.

Gene ontology enrichment analysis of transcripts with differential translational efficiencies during the first 60 min of WT vs injectisome mutant *Salmonella* infection showed they have functions related to cytokine activity and, strikingly, 75 transcription factors (Fig. 5D, S4D, Supplementary Data 3). In contrast, and similarly to previous transcriptomic studies[3], the transcriptional response was also enriched for cytokines and other cell signaling genes but there was no such enrichment for DNA-binding nor transcription-related genes. Beyond the initial 60 min, genes with injectisome-dependent transcriptional induction were also enriched for transcriptional functions, though the overlap with the rapidly translationally regulated transcription factors was small (Fig. S4E). Overall, this reinforces the hypothesis that rapid, injectisome-dependent translational induction of transcription modulators reshapes the transcriptional landscape and, consequently, the response of macrophages to *Salmonella* infection.

## Inducible blockage of the SPI-1 injectisome uncouples the impact of injectisome-membrane interactions from subsequently secreted effectors

*Salmonella* with intact SPI-1 injectisome triggers rapid translational and transcriptional induction of transcription factor synthesis in macrophages. These transcription factors can dictate infection outcomes, as in the case of *Egr1* where it significantly dampens the host inflammatory response. It is therefore of key importance to understand the mechanisms that drive such upregulation. We reasoned that both *Egr1* transcriptional stimulation and the potent translational induction (i.e., higher translation efficiency) could be a result of the direct interaction of the injectisome with the macrophage, or through insertion of the SipB/C translocon complex into the macrophage plasmalemma[21], or due to transport of effector proteins once the injectisome is fully assembled and is in a secretion-competent state. To distinguish between these possibilities, we engineered an injectisome-blocking substrate in which the effector protein SptP is C-terminally fused to the green fluorescent protein (SptP-GFP). This injectisome-blocking substrate is targeted to the export machinery and stalls within the injectisome export channel because the folding of the GFP moiety (unlike the effector protein sequence) is irreversible. Stalling occurs after needle assembly is complete, thereby blocking transport of effectors and translocon subunits, such as SipB and SipC, through the injectisomes that have assembled on the bacterial cell surface[74] (Fig. 6A, B). Inducible blockage of the injectisome, therefore, enables us to uncouple the effect of penetration of the macrophage plasmalemma from the effect of subsequently secreted effectors.

Expression of the SptP-GFP injectisome blocking substrate was controlled in the following manner: (1) uninduced, therefore recapitulating WT infection, where the injectisome penetrates macrophages and delivers effectors; (2) inducing expression of the blocking substrate 2 h prior to macrophage infection, resulting in assembly of injectisomes that are blocked with the SptP-GFP blocking substrate, preventing delivery of effectors whilst at the same time preventing delivery and insertion of translocon subunits (SipB and SipC) into the host cell membrane, which therefore abolishes penetration of the macrophage plasmalemma; and (3) blockage of injectisome induced at 5 min post infection, therefore allowing injectisomes to penetrate macrophage plasmalemma via the assembled translocon while inhibiting further delivery of effectors (Fig. 6A). This system was validated by assaying cytotoxicity with and without induction of the blocking substrate at 5 min post infection. As detailed above, the SPI-1 injectisome secretes effectors that are potent activators of the inflammasome, rapidly triggering host cell death (Fig. 1D). Expression of the blocking substrate at 5 min post infection prevents this host death, demonstrating that effector secretion into host cells has been successfully inhibited (Fig. 6C), without altering infection dynamics (Fig. S5).

As expected, the abundance of *Egr1* mRNA increases when macrophages were infected by *Salmonella* with unobstructed injectisomes, and expression was significantly impaired in macrophages challenged with *Salmonella* with pre-blocked, translocation-defective injectisomes. However, *Egr1* transcript accumulation was not significantly reduced when host cell membranes were penetrated by the injectisome but delivery of subsequent effectors was prevented by SptP-GFP induction at 5 min post infection (Fig. 6D). Taken together, these results suggest that the trigger for *Egr1* upregulation occurs very rapidly during infection and does not require transport of effectors into the host cell but rather is likely a result of injectisome penetration of the macrophage plasmalemma.

## The *Egr1* UTRs are sufficient to drive SPI-1-dependent translational upregulation

We next sought to understand the trigger governing the translational arm of *Egr1* upregulation. *Cis*-acting translation regulatory elements are commonly located in the untranslated regions (UTRs) of transcripts. To determine if such a feature conferring injectisome-dependent translational upregulation is present in the *Egr1* mRNA, a luminescence reporter system was established. Here, capped and polyadenylated mRNA encoding firefly luciferase and flanked by the

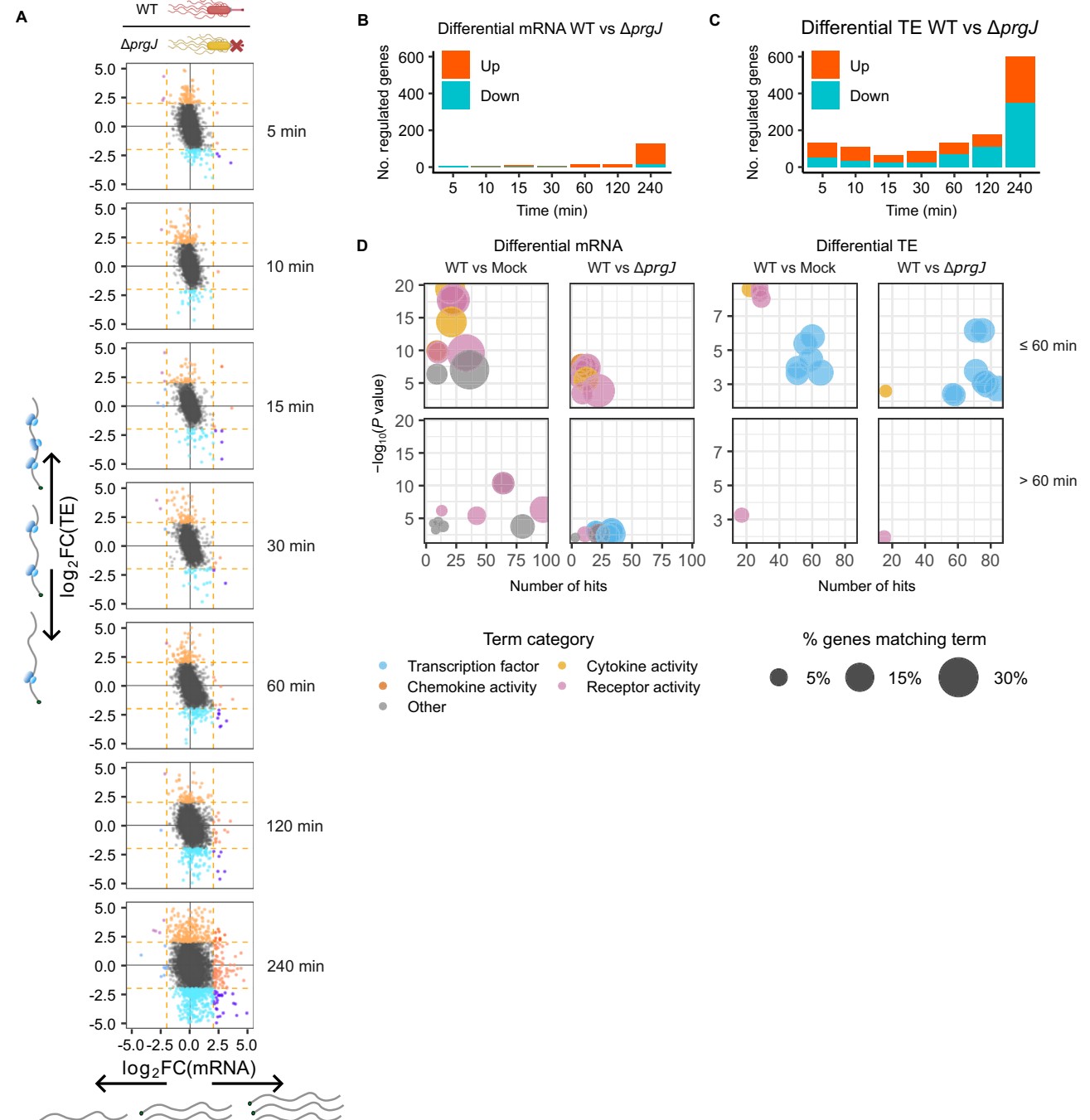

**Fig. 5 | Time-resolved Ribo-Seq reveals dynamic changes in host translation.** Data from two biological replicates. **A** Differential regulation of host gene expression represented as the log$_2$FC of translation efficiency (TE) versus log$_2$FC mRNA abundance in WT *Salmonella* infection compared to Δ*prgJ* infection. Dashed orange lines show the log$_2$FC cutoffs (±2) used to select differentially expressed genes, and genes that pass these thresholds are colored. Only genes with log$_2$FC in mRNA and TE between −5 and 5 are shown here (see Fig S4B for uncropped plots). Icons created in BioRender. Chung, B. (2025) https://BioRender.com/sjej7aq. Number of genes differentially expressed between WT and Δ*prgJ Salmonella* infection at both the TE (**B**) and mRNA (**C**) levels. **D** Top 10 enriched GO molecular function terms in differentially expressed genes at both the TE (right) and mRNA (left) levels. Genes were grouped by the timing of differential expression: at or before 60 min, and after 60 min post infection. *P* values were determined with Fisher's one-tailed test with Benjamini-Hochberg correction for multiple comparisons.

*Egr1* UTRs (Fig. 6E top), was transfected into cells. These cells were then infected with *Salmonella* where the SptP-GFP blocking substrate was either pre-induced or induced at 5 min post infection (Fig. 6A). Infection with *Salmonella* with unobstructed injectisome results in significant host cell death (Fig. 6C) which would confound assaying reporter expression. To identify differences in translational efficiency of the reporter between the infections, luciferase activity of cell lysates was assayed at 2 h post infection to align with peak EGR1 protein abundance (Fig. 3C, D). Luciferase activity was normalized by the amount of protein within the assayed lysate, and relative reporter mRNA abundance to account for differences in the rate of mRNA turnover. This system revealed that translational induction of *Egr1* is *Salmonella* injectisome penetration-dependent, effector-independent and is driven by the *Egr1* UTRs (Fig. 6E).

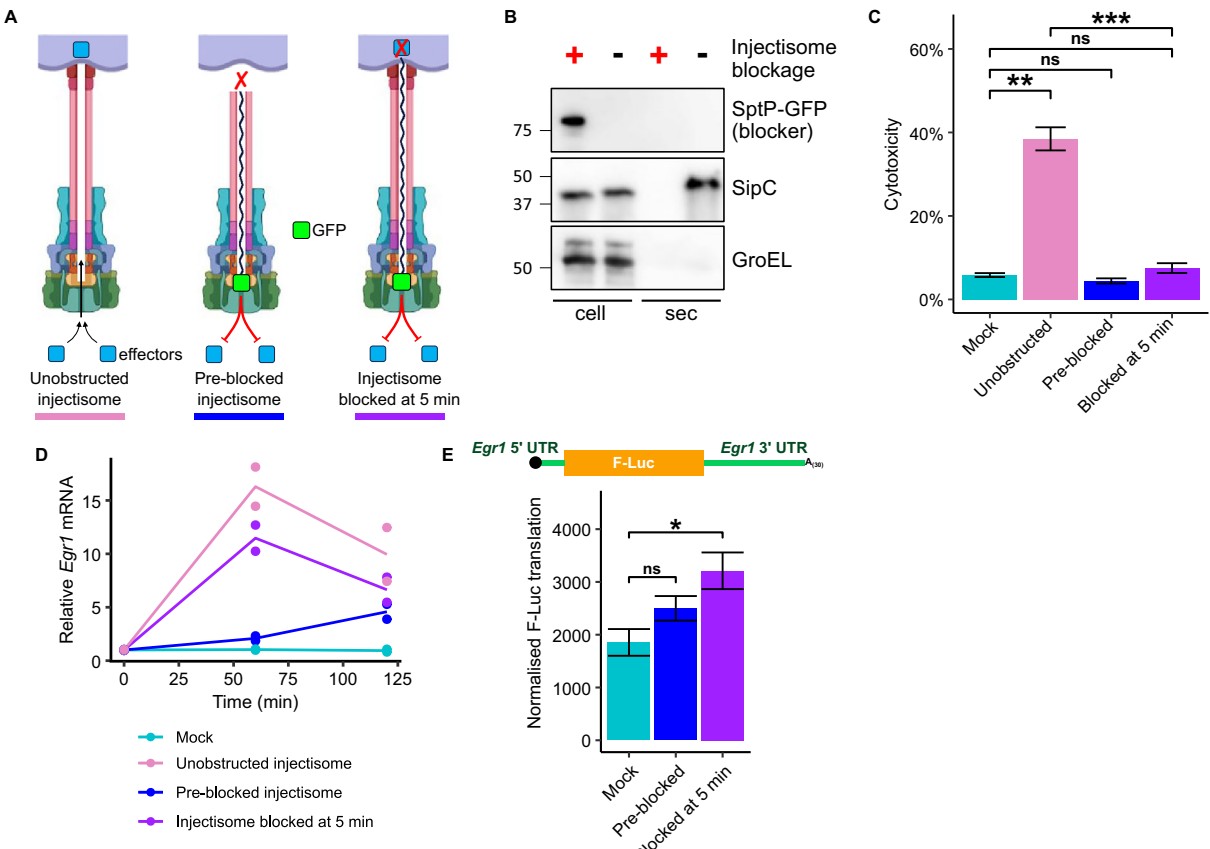

**Fig. 6 | Injectisome puncture of the host cell membrane triggers *Egr1* upregulation. A** Experimental outline illustrating the effect of the SptP-GFP injectisome blocking substrate on protein export via the injectisome. *Salmonella* cells not producing the blocking substrate can transport effector proteins into host cells (unobstructed injectisome, left). *Salmonella* cells expressing the injectisome blocking substrate before infection are unable to transport effector subunits via the injectisome (pre-blocked injectisome, middle). *Salmonella* cells were also incubated with macrophages for 5 min before inducing expression of the injectisome blocking substrate such that injectisomes can engage with macrophage cells but secretion of effectors proteins from *Salmonella* into macrophages is blocked at 5 min (injectisome blocked at 5 min, right). Created in BioRender. Chung, B. (2025) https://BioRender.com/sjej7aq. **B** Secretion analysis of WT *Salmonella* either expressing the SptP-GFP injectisome blocking subunit (+) or carrying the empty vector (-). Whole cell (cell) and secreted proteins (sec) from late-log-phase cultures were separated by SDS-PAGE and immunoblotted with anti-Myc-tag (SptP-GFP), anti-SipC, or anti-GroEL antisera. Representative of 3 independent experiments. **C** Cytotoxicity following infection with and without injectisome blockage as described in A. LDH concentration was determined in cell culture supernatants at 120 min post infection, LDH at 0 min was subtracted and values shown are relative to total cell lysis. Significance determined by one-sided Student's *t*-test with Benjamini-Hochberg adjusted *P* values of: Mock: Unobstructed = $1.0 \times 10^{-3}$, Mock: Pre-blocked = $9.4 \times 10^{-1}$, Mock: Blocked at 5 min = $1.7 \times 10^{-1}$, and Unobstructed: Blocked at 5 min = $9.5 \times 10^{-4}$; $n = 3$ biological replicates; data shows mean ± SEM. **D** *Egr1* transcript abundance across a *Salmonella* infection time course with blockage of the SPI-1 injectisome induced as indicated. **E** A capped, polyadenylated mRNA reporter (top) encoding firefly luciferase flanked by the *Egr1* 5′ and 3′ UTRs was transfected into cells, which was followed by *Salmonella* infection with blockage of the SPI-1 injectisome induced as indicated (MOI 10). Luciferase activity was assayed at 2 h, normalized by relative total protein and reporter mRNA abundance to determine the rate of reporter mRNA translation (bottom). Error bars show standard error of quotients; significance determined by two-sided Student's *t*-test with Benjamini-Hochberg adjusted *P* values of: Mock: Pre-blocked = $9.7 \times 10^{-2}$ and Mock: Blocked at 5 min = $3.0 \times 10^{-2}$; $n = 5$ biological replicates; error bars show mean ± SEM.

## Discussion

Cellular stress profoundly impacts gene expression dynamics[1–5,75,76]. Here, we demonstrate that *Salmonella* infection is a potent stressor that selectively induces host protein synthesis through modulating mRNA translation efficiency within the first hour of infection. This mechanism, primarily triggered by injectisome penetration of the macrophage plasmalemma rather than effector secretion, highlights the host cell's capacity to swiftly modulate translation in response to pathogen attack, complementing transcriptional response to biotic stress.

We showed that in macrophages, genes encoding DNA-binding transcription factor proteins, including *Egr1*, are translationally activated within first 60 min of injectisome penetration. *Egr1* was rapidly and transiently induced at both the transcriptional and translational levels, with mRNA and protein levels peaking at 60 and 120 min post infection respectively, before declining over the remaining timepoints. This rapid synthesis ensures EGR1 protein accumulation shortly after

injectisome interaction with the macrophage membrane. EGR1 was recently found to be involved in macrophage development by limiting the accessibility of inflammatory gene enhancers through recruitment of the NuRD chromatin-remodeling machinery[60]. EGR1 is also known to be rapidly and transiently induced in response to various stimuli and has a diverse range of roles, including regulating DNA replication and cell death[57,61,62,77]. We further revealed that, during *Salmonella* infection, EGR1 is a transcriptional suppressor that negatively regulates inflammation and cell death-associated genes such as *Il1b*. This tight temporal regulation enables *Egr1* to limit the inflammatory response and contributes to resistance against host cell death[78,79]. We hypothesize that this transient immunosuppression ultimately, and perhaps inadvertently, benefits *Salmonella*, allowing its persistence in surviving macrophages, as evidenced by most macrophages that survive injectisome-mediated infection continue to harbor viable bacteria. However, further in vivo studies will be required to confirm this.

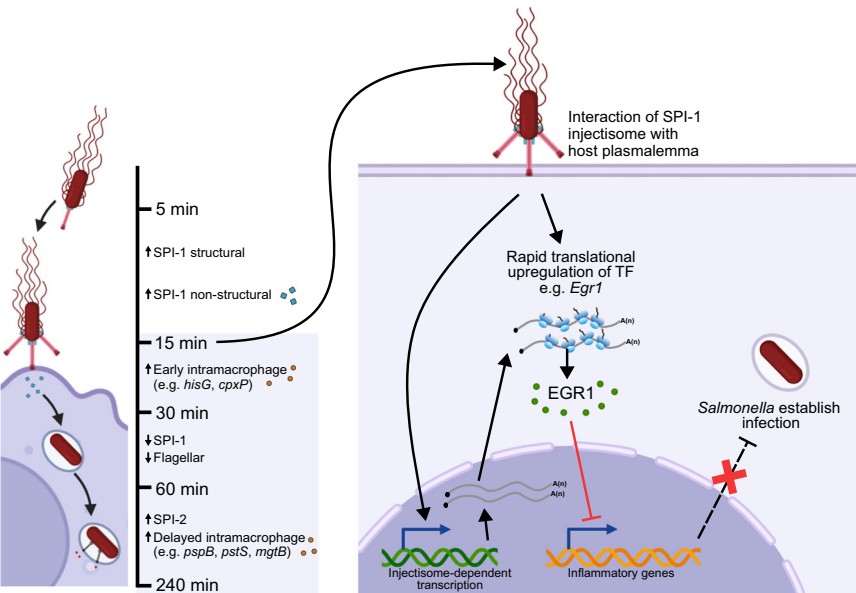

**Fig. 7 | Summary and model.** Specific classes of *Salmonella* genes are transiently expressed as infection is established, including rapid upregulation of SPI-1 genes as bacteria encounter host macrophages, leading to increased production of SPI-1 injectisomes as shown. *Salmonella* SPI-1 injectisome assembly forms transient pores in the host membrane which activates host signaling pathways, leading to increased *Egr1* transcription and enhanced translation. EGR1 negatively modulates immune gene transcription and, in doing so, inflammatory cytokine production in response to *Salmonella* infection is restrained, as is host cell death. Created in BioRender. Chung, B. (2025) https://BioRender.com/sjej7aq.

The surge of EGR1 protein abundance during injectisome-mediated infection is driven primarily by translational induction, mediated by the UTRs of the *Egr1* mRNA. These UTRs likely function as environmental sensors, responding to intracellular imbalances triggered by injectisome penetration. This mechanism enables swift EGR1 production, ensuring early modulation of host signaling pathways during infection. *Egr1* has previously been reported to be regulated by RNA binding proteins or through sequestration[80,81], though the importance of this in macrophages undergoing infection requires further investigation. Further investigation of UTR mediated translational regulation triggered in this way could uncover broader insights into pathogen exploitation of the host translation machinery and potential host counter-regulation during infection.

In summary, this study highlights the significance of rapid reprogramming of host gene expression during *Salmonella* infection, driven by injectisome penetration of the macrophage membrane. Upon encountering host macrophages, *Salmonella* swiftly boosts the expression of SPI-1 structural components, preparing for infection. Penetration of the macrophage membrane by the SPI-1 injectisome is a major trigger for the changes in host translation, which leads to rapid and robust transcription factor production. Amongst the translationally induced transcription factors is EGR1, a transcriptional suppressor of immune genes. Transient expression of EGR1 reshapes the host transcriptional landscape, restrains the inflammatory responses, and host cell death. We hypothesize that *Salmonella* exploits this brief period of immunosuppression to establish infection, leading to later downregulation of cytokine expression and host survival (Fig. 7). These findings illuminate a critical role of translational control in injectisome-mediated host–pathogen interactions, offering a deeper insight into bacterial exploitation of host cell gene expression machinery.

## Methods
### Mice
All mice were maintained in a specific pathogen-free facility according to the Animals Scientific Procedures outlined by the UK Home Office regulations. All work involving live animals complied with the University of Cambridge Ethics Committee regulations and was performed under the Home Office Project License numbers 80/2572 and P48B8DA35.

### Macrophage cell culture
Primary bone marrow derived macrophages (BMDM) were harvested as previously described[35]. Briefly, bone marrow from the rear legs of male 8-week-old C57BL/6 mice was extracted, suspended in culture media, and plated at a density of $10^6$ cells/ml supplemented with 20 ng/ml M-CSF (Peprotech). Cells were differentiated for 7 days, with additional M-CSF supplementation 4 days post extraction. Immortalized bone marrow-derived macrophages (iBMDM) were generated by retroviral transformation of bone marrow-derived macrophages as previously described[82]. Cells were routinely grown in DMEM supplemented with 10% fetal bovine serum at 37 °C and 5% $CO_2$.

### Salmonella infection
*Salmonella* Typhimurium SL1344 WT and $\Delta prgJ$ were sub-cultured in LB from stationary phase cultures and grown at 37 °C, with shaking at 200 rpm, until late exponential phase. Bacteria were washed and diluted in culture media, and added to cells at a multiplicity of infection of 10. For infections using primary bone marrow-derived macrophages, all media was supplemented with 20 ng/ml M-CSF. Unless otherwise indicated, after 15 min the media was supplemented with 100 μg/ml gentamicin. Mock samples were treated identically except an equivalent volume of media without bacteria was added.

### Microscopy
At the indicated timepoint, cells were fixed in 4% paraformaldehyde, permeabilized and stained with Phalloidin-CF594 conjugate (Biotium), and goat anti-*Salmonella* CSA-1 (Insight Biotechnology, #01-91-99) followed by anti-goat IgG conjugated to Alexa488 (Abcam, #ab150077). Cells were imaged and the proportion of infected cells was determined. Macrophages with intracellular bacteria, as determined by the intensity of CSA-1 staining and the position of *Salmonella* within the cell, were considered infected. Macrophages that were

uninfected (no intracellular bacteria) but had *Salmonella* associated with the host cell surface membrane were considered to be in the process of becoming infected.

## Gentamicin protection assay

The use of gentamicin protection assay to assess the number of intracellular bacteria has been described previously[2]. This protocol was modified to better account for host cell death. Briefly, 1 h post gentamicin treatment (i.e., 75 min post infection) macrophages were trypsinized, counted, and lysed in 0.09% Triton X-100. Serial dilutions of lysates were plated on LB-agar and grown at 37 °C overnight. The *Salmonella* colonies were counted and divided by the number of counted host macrophages to assess the number of colony forming units (CFU) per macrophage.

## Cytotoxicity assay

Cytotoxicity was measured by lactate dehydrogenase release using the CytoTox 96 Non-Radioactive Cytotoxicity Assay kit (Promega) according to the manufacturer's instructions. Cytotoxicity was determined relative to total cell lysis by 0.9% Triton X-100.

## Ribosome profiling with parallel RNA sequencing

Ribosome profiling was performed as previously described[38], a schematic of which is presented in Fig. 1E. Briefly, at the indicated time point, culture supernatant was removed from cells, cells were washed with PBS to remove dead and poorly attached cells, and the cell monolayer was flash frozen. Cells were lysed in buffer containing cycloheximide and chloramphenicol, and lysates were split for RNA-Seq and Ribo-Seq library preparation. For Ribo-Seq, lysates were treated with RNase I, and fragments protected from digestion by the ribosome were purified. For RNA-Seq, total cellular RNA was fragmented by alkaline hydrolysis. This was followed by library generation as previously described[38,83–85]. Sequencing was performed using NextSeq-500 or 2000 (Illumina).

Reads were aligned sequentially to mouse rRNA, mouse mRNA, *Salmonella* rRNA, and *Salmonella* mRNA. Mouse reference sequences were based on NCBI release mm10, and *Salmonella* reference sequences were based on GenBank sequences FQ312003.1, HE654725.1, HE654726.1, and HE654724.1. RiboSeqR[38] was used to confirm the quality of libraries and to count reads aligning to coding sequences. Read counts were normalized with edgeR[86] and edgeR was also used to filter *Salmonella* genes by expression for retention in further analysis. Xtail[87] was used to determine differential translational efficiency, and for the calculation and correction of the statistical significance. Gene set enrichment analysis was performed using g:Profiler[88].

## Western immunoblotting

Protein was harvested from cells disrupted with lysis buffer (50 mM Tris-HCl pH 8.0, 150 mM NaCl, 1 mM EDTA, 10% glycerol, 1% Triton X-100, 0.1% IGEPAL-CA630) that contained protease (cOmplete Mini, Roche) and phosphatase inhibitors (PhosSTOP, Roche). 30 µg of protein was separated by SDS-PAGE and transferred to a nitrocellulose membrane. EGR1 was detected using rabbit anti-EGR1 (Cell Signaling, #44D5) followed by anti-rabbit conjugated to HRP (Cell Signaling, #7074). The HRP signal was assayed using SuperSignal West Pico PLUS substrate (Thermo Scientific). GAPDH was detected using mouse anti-GAPDH (Sigma Aldrich, #G8795) followed by anti-mouse conjugated to IRDye 800 CW (Licor, #926-32210). SipC was detected by mouse anti-SipC (tgcBIOMICS, #tgc-a201-1) followed by anti-mouse IgG antibody conjugated to HRP (Promega, #W4021). GroEL was detected by rabbit anti-GroEL (Abcam, #ab90522) followed by anti-rabbit IgG antibody conjugated to HRP (Promega, #W4011). Myc tagged SptP-GFP blocking substrates were detected with anti-Myc HRP-conjugated mouse antibody (Cell Signaling, #2040).

## Reverse transcription quantitative polymerase chain reaction (RT-qPCR)

RNA was extracted from cells using TRIzol (Invitrogen) per the manufacturer's instructions. Reverse transcription was performed using M-MLV reverse transcriptase (Promega) with random hexamer primers per the manufacturer's instructions. Realtime qPCR was performed using iTaq Universal SYBR Green Supermix (Bio-Rad) and assayed on ViiA 7 system (Applied Biosystems) per the manufacturer's instructions. Primers were designed using PrimerBLAST (Supplementary Data 4).

## Knockout of *Egr1*

The Alt-R CRISPR-Cas system (IDT) was used per the manufacturer's instructions to edit the *Egr1* coding sequence using guide RNAs targeting either *Egr1* or no genes as a negative control (Supplementary Data 5)[89]. The system was delivered by lipofection into macrophages using Lipofectamine CRISPRMAX (Invitrogen). A clonal population was generated and targeted Sanger sequencing at the *Egr1* locus was performed (Genewiz) to confirm mutation of *Egr1*.

## mRNA 3′ end sequencing

Preparation of mRNA 3′ end sequencing libraries was performed using QuantSeq 3′ mRNA-Seq Library Prep Kit (Lexogen) with TRIzol extracted RNA and libraries were sequenced by Novogene Ltd. Reads were aligned to the mouse genome (mm10) and those aligning to genes counted. Read count normalization and differential expression analysis was performed using edgeR.

## Enzyme-linked immunosorbent assay (ELISA)

Culture supernatants were removed from infected cells at the indicated time point. ELISAs were performed to quantify IL-1β in these culture supernatants using the Mouse IL-1 beta/IL-1F2 DuoSet ELISA kit (R&D Systems) per the manufacturer's instructions.

## Blocked injectisome *Salmonella* transformant

Type III injectisomes can be blocked by fusing GFP to the C-terminus of an effector protein[74]. To generate our inducible blocking substrate construct, gDNA encoding the *Salmonella* chaperone SicP (residues 11–116) up to and including the downstream gene encoding the effector protein SptP (residues 1–543) was inserted into the pTrc99a plasmid[90] in-frame with the sequence encoding C-terminal GFP followed by a myc-tag. Isopropyl β-ᴅ-1-thiogalactopyranoside (IPTG) induction results in the production of an mRNA transcript encoding WT SicP chaperone which promotes efficient targeting of its cognate substrate (in this case the SptP-GFP-myc blocking substrate) to the injectisome export machinery. The mRNA also encodes the SptP-GFP-myc fusion protein which is targeted to the injectisome export machinery and stalls within the export channel, blocking the secretion of effector proteins via the SPI-1 injectisome. To block effector protein secretion via the injectisome, *Salmonella* cells carrying the inducible blocking substrate construct were grown in LB containing 100 µg/ml ampicillin and production of the blocking substrate (and the SicP chaperone) was achieved by supplementing the media with IPTG to a final concentration of 100 µM.

## Protein export assays

Export assays were performed as previously described[91]. Briefly, *Salmonella* strains were cultured at 37 °C in LB broth with 100 µM IPTG to mid-log phase (OD 1.5) for 2 h. Cells were centrifuged (6000 × *g*, 3 min) and resuspended in fresh media and grown for a further 60 min at 37 °C. The cells were pelleted by centrifugation (16,000 × *g*, 5 min) and the supernatant passed through a 0.2 µm nitrocellulose filter. Proteins were precipitated with 10% trichloroacetic acid (TCA) and 1% Triton X-100 on ice for 1 h, pelleted by centrifugation (16,000 × *g*, 10 min), washed with ice-cold acetone, and resuspended in SDS-PAGE loading

buffer (volumes calibrated according to cell densities). Fractions were analyzed by immunoblotting with anti-SipC (tgcBIOMICS), anti-Myc (Cell Signaling), and anti-GroEL (Abcam) anti-sera.

## Translational activation reporter assay

The firefly luciferase coding sequence, flanked by the UTRs of *Egr1* and with an $(A)_{30}$ tail, was cloned into the pTNT vector (Promega). In vitro transcription to generate reporter mRNA was performed from linearized plasmid using T7 polymerase with ARCA cap (NEB). Plasmid template was removed by DNase I digestion, and RNA was purified using acidic phenol-chloroform followed by precipitation in ethanol with ammonium acetate. The mRNA was further purified using Oligo Clean and Concentrator columns (Zymo Research) prior to transfection.

Purified mRNA was transfected into cells for 1 h using Lipofectamine MessengerMAX Transfection Reagent (Invitrogen). Cells were then infected or stimulated as indicated before being washed with PBS and lifted as for gentamicin protection assays. Cells from a single well were split for assaying luciferase activity and reporter mRNA abundance. To assess luciferase activity, the cells were pelleted and resuspended in Passive Lysis Buffer (Promega). Luciferase activity of lysates was assayed in the presence of D-luciferin (400 μM), $MgCl_2$ (1 mM), and ATP (50 μM) using a GloMax Navigator luminometer (Promega), two technical replicates each. Relative total protein was then quantified for each technical replicate using a Pierce BCA Protein Assay Kits (Thermo Scientific) and used to normalize luciferase activity by relative amount of total protein in lysates. RNA was harvested from the remaining cells by pelleting and resuspending in TRIzol. RT-qPCR was performed on extracted RNA as described above and relative luciferase activity was normalized by relative reporter mRNA abundance to determine relative reporter translation.

## Statistical analysis

Statistical analyses are detailed above. A list of statistical tests, *P* values and correction methods are detailed in Supplementary Data 7.

## Reporting summary

Further information on research design is available in the Nature Portfolio Reporting Summary linked to this article.

# Data availability

Raw and processed ribosome profiling and RNA sequencing data are available from the European Nucleotide Archive under study accessions ERP149959 and ERP179251 or can be found in the supplementary data. Source data are provided with this paper.

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

## Acknowledgements

We would also like to thank Jim Kaufman, Klaus Okkenhaug, and Alex Murphy for discussions and comments on the manuscript. G.W. was supported by the Department of Pathology PhD studentship and F.L. was supported by a BBSRC DTP studentship. PT and CEB were supported by Wellcome Trust (108045/Z/15/Z), B.Y.W.C., R.J., and M.B. and O.J.B. are supported by a Medical Research Council Fellowship to B.Y.W.C. [MR/R021821/1]. J.P. and M.B. are supported by a BBSRC project grant awarded to B.Y.W.C. [BB/V006096/1]. The B.Y.W.C. laboratory is supported by a Medical Research Council Fellowship [MR/R021821/1], BBSRC project grants [BB/X001261/1, BB/V017780/1, and BB/V006096/1] and a Royal Society Research Grant [RGS\R2\192222]. Figures created with BioRender.com.

## Author contributions

B.Y.W.C. conceived the research. B.Y.W.C., G.W., R.J., J.P., O.J.B., F.L., J.P.C., and C.E.B. designed experiments. R.J., J.P., and B.Y.W.C. generated RiboSeq, RNA-Seq, and QuantSeq libraries. G.W., R.J., and J.P. performed molecular and cell biology experiments. O.J.B. generated *Salmonella* mutants. F.L. performed bacterial bioinformatics. B.Y.W.C., G.W., R.J., J.P., F.L., and M.P.B. performed the bioinformatic analysis. C.E.B. provided macrophages and *prgJ* mutant. P.T. provided training for extracting BMDM. B.Y.W.C., G.W., O.J.B., and J.P.C. wrote the manuscript.

## Competing interests

The authors declare no competing interests.
