## [Transparent Peer review file · Nature Communications]

The Salmonella pathogenicity island 1 injectisome reprograms host cell translation to evade the inflammatory response

Corresponding Author: Dr Betty Chung

Version 0:

Reviewer comments:

Reviewer #1

(Remarks to the Author)

In the presented manuscript, Wood et al. investigate the reaction of macrophages to Salmonella infection, with a focus on translational regulation induced by the SPI-1 injectisome. Using RiboSeq, they measured changes in mRNA translation efficiencies upon infection of macrophages with wild-type or injectisome mutant Salmonella. They identified the mRNA encoding for the transcription factor EGR1 as more strongly activated at the level of translation in macrophages infected with wildtype Salmonella compared to macrophages infected with injectisome mutant Salmonella. By knocking out the *Egr1* gene, they demonstrate that EGR1 suppresses expression of inflammatory genes including *Il1b* and protects macrophages from cell death. In a separate RiboSeq experiment, they assess host and pathogen translation in parallel. Finally, they further validate injectisome-dependence of *Egr1* induction by blocking the injectisome before or briefly after infection.

Major points:

1. The authors aim to identify changes in host translation that depend on the SPI-1 injectisome. In Fig. S1B and Fig. 1C, they show that infection with mutant Salmonella is much less severe than with WT, so that after 15 min, the proportion of infected macrophages is four times lower for mutant Salmonella, and also colony forming units are strongly reduced after 75 min. When comparing fold-changes for mRNA abundance or translation efficiency upon infection in Fig. 1G, there seems to be a good correlation between WT and mutant infection, with overall weaker fold-changes in the mutant infection. This indicates that in general, mutant infection affects mRNA abundance and translational efficiency in a similar way as WT infection, just to a lesser extent, which could simply be explained by the decreased severity of infection by the mutant Salmonella. In this situation, interpretation of the data is challenging, because the general trend for a weaker reaction has to be taken into account (see scheme). This could be achieved by looking at the distance from a regression line that represents the general trend. Instead, the authors just set a fold-change cut-off of 1.5, which would mostly identify those genes with the strongest response upon infection in general. With this approach, *Il1b* translation seems differentially activated by injectisome-dependent infection, but for *Egr1* and *Nr4a1*, such an analysis is not possible because they could not be quantified in the mock infection condition. The analysis as it is presented by the authors convinces me that translation of these genes is affected by Salmonella infection, but a direct dependence on the action of the injectisome cannot be concluded.
2. In Fig. 4, the dependence of *Egr1* induction on the action of the injectisome is further investigated by blocking the injectisome. The reduction in cytotoxicity when the injectisome is blocked shows very well that the blockage works. When the blockage is performed before infection, translational induction of an F-Luc reporter carrying the *Egr1* UTRs is entirely abolished. This supports an injectisome-dependent regulation of *Egr1* translation. I still wonder whether this might be due to lower infection rates of the pre-blocked Salmonella. It would be good to see that the pre-blocked Salmonella are still able to infect macrophages to a similar degree as the injectisome-mutant Salmonella in Fig. 1.
3. The RiboSeq time-course presented in Fig. 3D shows negative correlations between changes in TE and changes in mRNA abundance. The authors should not leave this uncommented. There can be different reasons for such an observations, like spurious correlations (see Larsson et al., PNAS 2010, PMID: 21115840), or a temporal lag between

mRNA abundance and footprint, which is especially relevant when gene expression is far away from steady-state (see Schott et al., Nature Methods 2021, PMID: 34480152). In any case, it is problematic to conclude that translation efficiency is differentially regulated just from TE fold-changes in this case.

4. Concerning statistical analyses, I noted that several experiments contain only two replicates. While it is possible to calculate standard deviations and perform t-tests on only $n = 2$, statistical power is not great. For Fig. 4D, the authors emphasize in the text that the difference between unobstructed injectisome and injectisome blocked after 5 min is not significant, implying that there is no difference. In Fig. 4E, however, no statistical testing is performed at all, perhaps because “Blocked at 5 min” versus “Mock” would not be significantly different, too? In my eyes, this experiment is rather crucial to make the point that Egr1 translation is regulated in an injectisome-dependent manner, and would benefit from more replicates to have sufficient statistical power. In analyzing the RiboSeq data, it seems that the authors relied exclusively on fold-changes, and no statistical criteria were applied at all, which I found surprising. Standard tools like EdgeR or DESeq2, however, would not allow to take general trends into account, as I suggested above. It is also not clear to me how many replicates were performed for the RiboSeq and RNASeq experiments. I was not able to find this information in the figure legends.

5. There is no mechanistical insight into the regulation of Egr1 at all. It has been described that Egr1 mRNA has an AU-rich element in its 3'UTR, is translationally regulated by HuR (PMID: 23127556) and recruited to Stress Granules (PMID: 38940347). These possible mechanisms are not investigated or discussed.

Minor points:

6. Line 107: the word “showed” does not make sense here
7. The sentence in line 188 – 189 seems to be truncated.
8. On page 10, the reference to Figure 3B seems to be wrong.
9. Line 385: “uncouples”
10. Line 473: “as evidenced by the fact that”?
11. Legend of Fig. 2: Panel H to L should be E to I?

Reviewer #2

(Remarks to the Author)

Wood, Johnson et al. present a manuscript that carefully analyses bacterial and host transcriptional and translational responses upon infection of macrophages with the Salmonella wild type or a Δ prgJ (T3SS-1 neg.) mutant using RNA seq and Ribo seq over time. The study thus provides a wealth of new information on the in vitro host-pathogen interaction of Salmonella and BMDMs.

The study is split in two parts: First, the identification and validation of genes that are regulated on the translational level 60 min post infection of immortalized BMDMs with a particular focus on the transcription factor Egr1. Second, the time-resolved analysis of transcription and translation dynamics of infecting bacteria and infected host cells up to 240 min post infection using primary BMDMs.

The first part identifies genes that are not only transcriptionally upregulated in a T3SS-1-dependent manner but on top of that also show an enhanced translation efficiency. The authors hypothesize that this supports the very rapid response of the iBMDMs to infection with Salmonella. They validate that the upregulation of the transcription factor Egr1 restrains the inflammatory response of iBMDMs in response to infection by Salmonella.

The authors performed an experiment using a T3SS-1-blocking substrate to distinguish whether the T3SS-1-dependent effect of infection on Egr1 upregulation depended on the insertion of the T3SS translocon pore into the plasma membrane or on effector injection. While they could unsurprisingly show that blocking the T3SS-1 from the start led to the absence of Egr1 mRNA upregulation, inducing the T3SS-1-blocking substrate only 5 min post infection did not show a significant difference in Egr1 upregulation. The authors concluded from this result that the translocon pore formation leads to the observed. I feel that this conclusion may not be justified given the experimental set-up. It takes a few minutes from the addition of an inducer (here IPTG) to reach relevant levels of the blocking substrate. I would assume that the Salmonella have at least 10 min post infection to inject effector proteins until the blocking substrate can do its job. In these 10 min, a sufficient amount of effector proteins might already have been injected to lead to Egr1 upregulation, so that one cannot necessarily conclude that the translocon formation is responsible for this effect. The authors could control the dynamics of T3SS-1 blocking by testing secretion in vitro using more sensitive enzyme-based T3SS secretion assays. In addition, I suggest the use of Δ effector mutants and/or Δ class I chaperone mutants (e.g. Δ sicP, Δ invB) to distinguish between the effect of translocon formation and injected effectors.

Overall, the paper presents a very thorough analysis and the study contributes plenty of data that will be received well by the community.

Minor points:

Line 61: To call SipBCD effectors is unusual. Better would be the term ‘tip and translocon components’

Line 63: Ref 10 is not correct here. Marlovits 2006 may be better. Also, PrgJ does not form a channel as originally suggested but just forms a single layer adapter (Torres-Vargas Mol Micro 2019, Hu Nature Microbiol 2019).

Line 332: What does 'proximity' mean in this context. I am not sure if this is the correct wording.

Reviewer #3

(Remarks to the Author)

Wood et al use an interesting combination of proteomics and transcriptomics to assess how early interactions between invasive *Salmonella* and host macrophages are shaped by a balance between invasion-mediating bacterial secretion systems and host cell's translational responses. The role of the *Salmonella* SPI-1 T3SS in triggering host cell death in macrophage infection has been established *ex vivo*, but it has remained unclear how the balance of host-pathogen interactions skews these factors towards promotion of infection. Here, the authors describe how rapid translation of host factor EGR1 contributes to a transient anti-inflammatory state that paradoxically enhances bacterial infection. The data suggest that this is attributable at least in part to control at the translational level. These data provide new advances in understanding the trade-off between host cell factors that limit infection and bacterial activities that subvert these responses, and provide interesting insights into the temporal dynamics of host cell responses to bacterial invasion and early-stage intracellular activities. The paper is very well written and covers an impressive range of omics experiments. By addressing a few points detailed below, the authors may want to further strengthen the impact of their work.

Main comments

1. Fig.1 E-K; Fig. 2A-D: the authors may want to explain more clearly how they accounted for different invasion rates of wt vs prgJ. How could this affect the triggering kinetics of cell-intrinsic immunity with possible downstream effects on transcription or translation of Egr1 or other genes? Similarly, does GFP-blocker expression slow down the macrophage invasion kinetics in Fig. 4 experiments?
2. Fig. 2. The authors may want to explain more clearly if *Salmonella* shows the same invasion efficiency in the Egr1 KO cells compared to the wild type.
3. In absence of host infection data, the authors may want to tone down the statement about the benefits for the pathogen (lines 472-474).

Minor comments:

4. Line 66: The authors may wish to explain at this stage that *ex vivo* macrophage invasion via SPI-1 T3SS has been studied previously (e.g. PMID: 31113898; PMID: 16445685). The invasion kinetics will become important for interpreting the presented data.
5. Fig S1B vs Fig S3A, lines 309-316: the infection rate of the prgJ mutant in iBMDMs (25%) vs the rate in primary macrophages (50%) is interesting, especially given the infection rate for WT is 75% in both cell types. The authors briefly state this (lines 314-316), but the readers might be interested in a more detailed discussion?
6. Lines ~331-353: various references to Fig 3B should I think refer to Fig 3C, perhaps this is an error?

Reviewer #4

(Remarks to the Author)

Version 1:

Reviewer comments:

Reviewer #1

(Remarks to the Author)

My first two comments referred to the problem that WT, injectisome-mutant and -blocked *Salmonella* infect macrophages to different degrees. The authors have now provided data showing that infection rates are similar between injectisome-mutant and -blocked *Salmonella* (Fig. S5). Regarding the difference to WT, which was also raised as an issue by Reviewer #3, the authors convinced me to leave the interpretation to the reader.

Regarding point 3, I do not agree that the negative correlations can simply be ignored. Despite of the strict fold-change cut-offs set by the authors, it seems that the majority of mRNAs with a log₂ TE fold-change > 2 has a negative mRNA log₂ fold-change, while the majority of mRNAs with a log₂ fold-change < -2 has a positive mRNA log₂ fold-change, especially in time-points up to 120 min. This indicates that either mRNA abundance and translation efficiency are actively regulated in opposite directions between WT and injectisome-mutant for many mRNAs, or that there is some artefact.

Regarding my comments on statistical analyses, additional replicates were added for Fig. 4E. I still cannot find the information about the number of replicates for the Ribo-seq experiments presented in Fig. 1 and 3 in the figure legends, the Methods section (lines 552 – 571) or the main text describing these results. I would recommend to mention the number of replicates for all experiments in the respective figure legends, where they can be found more easily.

No experiments were added to further elucidate the mechanism of Egr1 translational regulation, e.g. by its AU-rich element, recruitment to Stress Granules or binding to HuR, which leaves this part rather weak on the mechanistic side. The authors added three lines to their discussion to mention these possibilities.

All my minor suggestions for corrections were taken into account.

Reviewer #2

(Remarks to the Author)

The authors have responded well to the reviewers comments, which improved the manuscript substantially. I have no further issues to raise.

Reviewer #3

(Remarks to the Author)

The authors nicely have addressed all my comments.

Reviewer #4

(Remarks to the Author)

REVIEWER COMMENTS

Reviewer #1 (Remarks to the Author):

In the presented manuscript, Wood et al. investigate the reaction of macrophages to Salmonella infection, with a focus on translational regulation induced by the SPI-1 injectisome. Using RiboSeq, they measured changes in mRNA translation efficiencies upon infection of macrophages with wild-type or injectisome mutant Salmonella. They identified the mRNA encoding for the transcription factor EGR1 as more strongly activated at the level of translation in macrophages infected with wildtype Salmonella compared to macrophages infected with injectisome mutant Salmonella. By knocking out the Egr1 gene, they demonstrate that EGR1 suppresses expression of inflammatory genes including Il1b and protects macrophages from cell death. In a separate RiboSeq experiment, they assess host and pathogen translation in parallel. Finally, they further validate injectisome-dependence of Egr1 induction by blocking the injectisome before or briefly after infection.

Major points:

1. The authors aim to identify changes in host translation that depend on the SPI-1 injectisome. In Fig. S1B and Fig. 1C, they show that infection with mutant Salmonella is much less severe than with WT, so that after 15 min, the proportion of infected macrophages is four times lower for mutant Salmonella, and also colony forming units are strongly reduced after 75 min. When comparing fold-changes for mRNA abundance or translation efficiency upon infection in Fig. 1G, there seems to be a good correlation between WT and mutant infection, with overall weaker fold-changes in the mutant infection. This indicates that in general, mutant infection affects mRNA abundance and translational efficiency in a similar way as WT infection, just to a lesser extent, which could simply be explained by the decreased severity of infection by the mutant Salmonella. In this situation, interpretation of the data is challenging, because the general trend for a weaker reaction has to be taken into account (see scheme). This could be achieved by looking at the distance from a regression line that represents the general trend. Instead, the authors just set a fold-change cut-off of 1.5, which would mostly identify those genes with the strongest response upon infection in general. With this approach, Il1b translation seems differentially activated by injectisome-dependent infection, but for Egr1 and Nr4a1, such an analysis is not possible because they could not be quantified in the mock infection condition. The analysis as it is presented by the authors convinces me that translation of these genes is affected by Salmonella infection, but a direct dependence on the action of the injectisome cannot be concluded.

We appreciate the reviewers thoughtful and detailed comments. However, we would disagree that the identification of up regulated genes may represent a common response to infection, independent of the injectisome.

The reviewer describes the injectisome mutant as driving a weaker response than WT as compared to the mock, however many of the genes identified in Fig. 1G-H show an equal or greater response in the injectisome mutant compared to WT.

The fact that most genes share a similar trend is not unsurprising given the known important role of cell surface receptors such as TLR4 in detecting bacterial pathogens and driving a translational response. *Tnf* is the prototypical TLR4 upregulated gene, which we identify as an example with robust upregulation in both infections in Fig. 1G. *Tnf* would be an ideal fit for scenario described by the reviewer, i.e. low levels in uninfected cells with rapid transcriptional and translational upregulation, but it is not selected in our analyses. Rather, we are able to distinguish the general response to infection against the response dependent on the *Salmonella* injectisome and it is in this comparison that we identified *Egr1*, *Nr4a1* and *Il1b*.

2. In Fig. 4, the dependence of *Egr1* induction on the action of the injectisome is further investigated by blocking the injectisome. The reduction in cytotoxicity when the injectisome is blocked shows very well that the blockage works. When the blockage is performed before infection, translational induction of an F-Luc reporter carrying the *Egr1* UTRs is entirely abolished. This supports an injectisome-dependent regulation of *Egr1* translation. I still wonder whether this might be due to lower infection rates of the pre-blocked *Salmonella*. It would be good to see that the pre-blocked *Salmonella* are still able to infect macrophages to a similar degree as the injectisome-mutant *Salmonella* in Fig. 1.

We entirely agree that by comparing the number intracellular bacteria upon infection with *Salmonella* with blocked injectisomes to injectisome-mutant *Salmonella* will give reassurance that the *Egr1* upregulation is not due differences in infection rates. We have added Fig S5 comparing intracellular bacteria number in infection the blocking strains. This demonstrated in infection the pre-blocked *Salmonella* are internalised at 0.35 CFU per macrophage, comparable to the injectisome mutant in Fig. 1C. Notably, we saw comparable internal bacteria counts for infection where blockage was left uninduced and induced at 5 min. To confirm increased *Egr1* expression is not simply due to increased numbers of intracellular bacteria, we infected macrophages with *Listeria*, which invade through the zipper mechanism, i.e. not using a T3SS. *Egr1* expression in infection with cytoplasmic or vacuolar (Δhly) *Listeria* was significantly lower than WT *Salmonella*, Fig. R1, and comparable to $\Delta prgJ$ *Salmonella* in Fig. 2B. It is also important to note that, as described in lines 238-239, *Egr1* induction has been observed in other bacteria:host cell type interactions to depend on a T3SS, including those that do not enter the host cell such as EPEC (PMID: 11553563).

Figure R1: Relative *Egr1* expression at 60 min post infection of iBMDMs with WT *Salmonella*, WT *Listeria* or vacuolar trapped (Δhly) *Listeria*.

3. The RiboSeq time-course presented in Fig. 3D shows negative correlations between changes in TE and changes in mRNA abundance. The authors should not leave this uncommented. There can be different reasons for such an observations, like spurious correlations (see Larsson et al., PNAS 2010, PMID: 21115840), or a temporal lag between mRNA abundance and footprint, which is especially relevant when gene expression is far away from steady-state (see Schott et al., Nature Methods 2021, PMID: 34480152). In any case, it is problematic to conclude that translation efficiency is differentially regulated just from TE fold-changes in this case.

There is a slight negative correlation in Fig. 3D. The slight correlation is present in all time points, suggesting a minimal role of temporal lag between transcription and translation. In addition, the magnitude of differences is considerably greater for TE than mRNA. “Spurious correlation” arises where differences in mRNA levels may be interpreted as differences in translational efficiency due to TE being calculated with mRNA levels as a denominator (Fig. 1F), this is particularly impactful when calculating p values without looking at the underlying data. The graphs in Fig. 3D show the log₂ foldchange in TE and mRNA between the indicated comparisons and genes with differential mRNA and differential TE were selected using fold change cutoffs. The use of foldchange cutoffs ensures a minimal impact of spurious correlation on the selection of differentially regulated genes. It should also be noted that the greatest variation in all panels is along the orthogonal axes, indicating differential TE and mRNA values do not harbour enough covariation to impact the bulk interpretation from this data.

4. Concerning statistical analyses, I noted that several experiments contain only two replicates. While it is possible to calculate standard deviations and perform t-tests on only $n = 2$, statistical power is not great. For Fig. 4D, the authors emphasize in the text that the difference between unobstructed injectisome and injectisome blocked after 5 min is not significant, implying that there is no difference. In Fig. 4E, however,

no statistical testing is performed at all, perhaps because “Blocked at 5 min” versus “Mock” would not be significantly different, too? In my eyes, this experiment is rather crucial to make the point that Egr1 translation is regulated in an injectisome-dependent manner, and would benefit from more replicates to have sufficient statistical power.

We have revisited the experimental setup used in Fig. 4E. Previously, parallel wells were analysed for relative luciferase activity and relative reporter mRNA abundance. Now, cells from a single well are split for RNA and protein harvest, allowing the two values to be directly tied together for an individual replicate. Using this setup, we have repeated the experiment with more samples ($n=5$), giving us the statistical power to demonstrate the relative luciferase activity in the “Blocked at 5 min” condition versus “Mock” is significantly greater, adjusted p -value = 0.03.

In analysing the RiboSeq data, it seems that the authors relied exclusively on fold-changes, and no statistical criteria were applied at all, which I found surprising. Standard tools like EdgeR or DESeq2, however, would not allow to take general trends into account, as I suggested above. It is also not clear to me how many replicates were performed for the RiboSeq and RNASeq experiments. I was not able to find this information in the figure legends.

The RiboSeq/RNASeq time course data were generated from two independent experiments, as detailed in the Methods section. For normalization, both EdgeR and Xtail (PMID: 27041671) were used, in combination with stringent filtering criteria specific to RiboSeq data (PMID: 26286745, 32284544). These measures ensured that the analysis focused exclusively on genes with robust evidence of active translation. All genes discussed exhibit statistically significant changes in translational efficiency (TE), supported by both computational analysis and direct visual inspection (e.g., Figure 1I and K). The application of fold change thresholds to

identify differentially expressed genes is standard practice in translomics (e.g., PMID: 39169219), and all reported genes meet these criteria.

5. There is no mechanistical insight into the regulation of Egr1 at all. It has been described that Egr1 mRNA has an AU-rich element in its 3'UTR, is translationally regulated by HuR (PMID: 23127556) and recruited to Stress Granules (PMID: 38940347). These possible mechanisms are not investigated or discussed.

The reporter experiment shown in Figure 4E was designed to test whether the injectosome-dependent induction of Egr1 translation is mediated by its UTRs. Notably, a significant increase in translation was observed only under the condition where the injectosome was blocked at 5 minutes. This suggests that the observed effect is driven by cis-acting elements within the UTRs, as discussed in the manuscript. In response to the reviewer's suggestion, we have expanded this discussion (lines 480–482) to include previous reports implicating RNA-binding proteins and RNA sequestration mechanisms in the regulation of Egr1 expression.

Minor points:

6. Line 107: the word “showed” does not make sense here

We thank the reviewer for identifying this and have edited this line for clarity.

7. The sentence in line 188 – 189 seems to be truncated.

We thank the reviewer for identifying this and have edited this line for clarity.

8. On page 10, the reference to Figure 3B seems to be wrong.

We thank the reviewer for identifying the incorrectly referenced figure and have corrected it to Figure 3C.

9. Line 385: “uncouples”

We thank the reviewer for identifying this typo and have corrected it.

10. Line 473: “as evidenced by the fact that”?

We thank the reviewer for identifying this and have clarified the wording.

11. Legend of Fig. 2: Panel H to L should be E to I?

We thank the reviewer for identifying this error and have updated the legend of Figure 2 to reference the correct panels.

Reviewer #2 (Remarks to the Author):

Wood, Johnson et al. present a manuscript that carefully analyses bacterial and host transcriptional and translational responses upon infection of macrophages with the *Salmonella* wild type or a Δ prgJ (T3SS-1 neg.) mutant using RNA seq and Ribo seq over time. The study thus provides a wealth of new information on the in vitro host-pathogen interaction of *Salmonella* and BMDMs.

The study is split in two parts: First, the identification and validation of genes that are regulated on the translational level 60 min post infection of immortalized BMDMs with a particular focus on the transcription factor Egr1. Second, the time-resolved analysis of transcription and translation dynamics of infecting bacteria and infected host cells up to 240 min post infection using primary BMDMs.

The first part identifies genes that are not only transcriptionally upregulated in a T3SS-1-dependent manner but on top of that also show an enhanced translation efficiency. The authors hypothesize that this supports the very rapid response of the iBMDMs to infection with *Salmonella*. They validate that the upregulation of the transcription factor Egr1 restrains the inflammatory response of iBMDMs in response to infection by *Salmonella*.

The authors performed an experiment using a T3SS-1-blocking substrate to distinguish whether the T3SS-1-dependent effect of infection on Egr1 upregulation depended on the insertion of the T3SS translocon pore into the plasma membrane or on effector injection. While they could unsurprisingly show that blocking the T3SS-1 from the start led to the absence of Egr1 mRNA upregulation, inducing the T3SS-1-blocking substrate only 5 min post infection did not show a significant difference in Egr1 upregulation. The authors concluded from this result that the translocon pore formation leads to the observed. I feel that this conclusion may not be justified given the experimental set-up. It takes a few minutes from the addition of an inducer (here IPTG) to reach relevant levels of the blocking substrate. I would assume that the *Salmonella* have at least 10 min post infection to inject effector proteins until the blocking substrate can do its job. In these 10 min, a sufficient amount of effector proteins might already have been injected to lead to Egr1 upregulation, so that one cannot necessarily conclude that the translocon formation is responsible for this effect. The authors could control the dynamics of T3SS-1 blocking by testing secretion in vitro using more sensitive enzyme-based T3SS secretion assays. In addition, I suggest the use of Δ effector mutants and/or Δ class I chaperone mutants (e.g. Δ sicP, Δ invB) to distinguish between the effect of translocon formation and injected effectors.

While using Δ effector mutants is an appealing proposal, we foresee a fundamental flaw. That is, structural components of the injectisome have effector functions, e.g. SipC has roles in host membrane protein trafficking (PMID: 27078059) and bundling F-actin (PMID: 20212042), or are detected by the host cell, for example PrgJ activates the inflammasome. There is also some promiscuity in effector secretion,

with, for example, flagellin reported to be secreted through the SPI-1 injectisome (PMID: 17911114), where it can also activate the inflammasome. It is therefore not possible to negate the injectisome dependent host-pathogen interactions using simple deletions of all potential proteins secreted by the SPI-1 injectisome, which is what led us to employ the blocking substrate.

To validate inducing injectisome blockage at 5 min successfully limits effector secretion, we utilised a host cytotoxicity assay. Pyroptosis is induced following the highly sensitive detection of injectisome-secreted proteins including PrgJ and flagellin (PMID 29180436) with dramatic signal amplification (PMIDs 26449475, 26449474). With no increase in cell death following infection with the blocker induced at 5 min, we feel confident that effector secretion into the host cell has been successfully blocked. This therefore fulfils the reviewers request for a sensitive enzymatic T3SS secretion assay within the infection context.

Overall, the paper presents a very thorough analysis and the study contributes plenty of data that will be received well by the community.

Minor points:

Line 61: To call SipBCD effectors is unusual. Better would be the term 'tip and translocon components'

We initially referred to them as effectors as, in addition to their structural roles, they have functions as effectors e.g. SipC has roles in host membrane protein trafficking (PMID: 27078059) and bundling F-actin (PMID: 20212042). However, for clarity we have incorporated the suggestion to refer to them as tip and translocon components at this point.

Line 63: Ref 10 is not correct here. Marlovits 2006 may be better. Also, PrgJ does not not form a channel as originally suggested but just forms a single layer adapter (Torres-Vargas Mol Micro 2019, Hu Nature Microbiol 2019).

Miletic 2021 (Reference 10) is a publication showing the bacterial injectisome structure in high detail, including the position of PrgJ within it. We have added the additional reference to Marlovits 2006 and clarified our description of the function of PrgJ within the injectisome structure as acting as an adapter that connects the channel down which effectors are secreted.

Line 332: What does 'proximity' mean in this context. I am not sure if this is the correct wording.

Salmonella rapidly induce expression following addition to the macrophage culture, as compared to broth. While this may be physical contact, it may also be sensing of other factors originating from the macrophages and therefore we believe proximity is

the most accurate word to describe this. We have reworded this section to “*Salmonella* responds to being in proximity to the macrophages” to clarify this.

Reviewer #3 (Remarks to the Author):

Wood et al use an interesting combination of proteomics and transcriptomics to assess how early interactions between invasive *Salmonella* and host macrophages are shaped by a balance between invasion-mediating bacterial secretion systems and host cell's translational responses. The role of the *Salmonella* SPI-1 T3SS in triggering host cell death in macrophage infection has been established *ex vivo*, but it has remained unclear how the balance of host-pathogen interactions skews these factors towards promotion of infection. Here, the authors describe how rapid translation of host factor EGR1 contributes to a transient anti-inflammatory state that paradoxically enhances bacterial infection. The data suggest that this is attributable at least in part to control at the translational level. These data provide new advances in understanding the trade-off between host cell factors that limit infection and bacterial activities that subvert these responses, and provide interesting insights into the temporal dynamics of host cell responses to bacterial invasion and early-stage intracellular activities. The paper is very well written and covers an impressive range of omics experiments. By addressing a few points detailed below, the authors may want to further strengthen the impact of their work.

Main comments

1. Fig.1 E-K; Fig. 2A-D: the authors may want to explain more clearly how they accounted for different invasion rates of wt vs prgJ. How could this affect the triggering kinetics of cell-intrinsic immunity with possible downstream effects on transcription or translation of Egr1 or other genes?

The bacteria are added at the same MOI, thereby stimulating the cell surface receptors such as TLR4 equally in both infections, which is a major driver of the response to *Salmonella*. That said, WT *Salmonella* are able to invade via the trigger mechanism in addition to phagocytosis, thereby resulting in greater proportion of cells harbouring intracellular bacteria. To uncouple injectisome penetration from effector secretion, we performed the experiments outlined in Fig. 4.

Similarly, does GFP-blocker expression slow down the macrophage invasion kinetics in Fig. 4 experiments?

Fig S5 is added which shows infection with *Salmonella* expressing the blocking substrate results in a comparable number of intracellular bacteria as the injectisome mutant.

2. Fig. 2. The authors may want to explain more clearly if Salmonella shows the same invasion efficiency in the *Egr1* KO cells compared to the wild type.

Fig S2B is added to show similar numbers of intracellular bacteria in both negative control *EGR1*^{WT} and *EGR1*^{KO} macrophages, indicating equal susceptibility to bacterial invasion.

3. In absence of host infection data, the authors may want to tone down the statement about the benefits for the pathogen (lines 472-474).

We propose a mechanism whereby the downregulation of immune related genes triggered by *Egr1* upregulation promotes the bacteria's ability to establish infection. We have added to line 475 to state that further in vivo studies would be needed to confirm this.

Minor comments:

4. Line 66: The authors may wish to explain at this stage that ex vivo macrophage invasion via SPI-1 T3SS has been studied previously (e.g. PMID: 31113898; PMID: 16445685). The invasion kinetics will become important for interpreting the presented data.

Further discussion of the previously studied importance of the injectisome in macrophage invasion is added to line 66-67.

5. Fig S1B vs Fig S3A, lines 309-316: the infection rate of the prgJ mutant in iBMDMs (25%) vs the rate in primary macrophages (50%) is interesting, especially given the infection rate for WT is 75% in both cell types. The authors briefly state this (lines 314-316), but the readers might be interested in a more detailed discussion?

We have added a more detailed discussion at lines 317-318

6. Lines ~331-353: various references to Fig 3B should I think refer to Fig 3C, perhaps this is an error?

We thank the reviewer for identifying this error and have corrected it.

Reviewer #4 (Remarks to the Author):

Reviewer #1 (Remarks to the Author):

My first two comments referred to the problem that WT, injectisome-mutant and -blocked Salmonella infect macrophages to different degrees. The authors have now provided data showing that infection rates are similar between injectisome-mutant and -blocked Salmonella (Fig. S5). Regarding the difference to WT, which was also raised as an issue by Reviewer #3, the authors convinced me to leave the interpretation to the reader.

Regarding point 3, I do not agree that the negative correlations can simply be ignored. Despite of the strict fold-change cut-offs set by the authors, it seems that the majority of mRNAs with a log₂ TE fold-change > 2 has a negative mRNA log₂ fold-change, while the majority of mRNAs with a log₂ fold-change < -2 has a positive mRNA log₂ fold-change, especially in time-points up to 120 min. This indicates that either mRNA abundance and translation efficiency are actively regulated in opposite directions between WT and injectisome-mutant for many mRNAs, or that there is some artefact.

Regarding my comments on statistical analyses, additional replicates were added for Fig. 4E. I still cannot find the information about the number of replicates for the Ribo-seq experiments presented in Fig. 1 and 3 in the figure legends, the Methods section (lines 552 – 571) or the main text describing these results. I would recommend to mention the number of replicates for all experiments in the respective figure legends, where they can be found more easily.

No experiments were added to further elucidate the mechanism of Egr1 translational regulation, e.g. by its AU-rich element, recruitment to Stress Granules or binding to HuR, which leaves this part rather weak on the mechanistic side. The authors added three lines to their discussion to mention these possibilities.

All my minor suggestions for corrections were taken into account.

We thank the reviewer for their kind comments. Regarding the slight anticorrelation observed in Fig. 3D (now Fig. 5D). Discussion of the possible influence of the lag between transcription and translation may be responsible for the anti-correlation, and therefore the importance of using robust fold change cut-offs, is included in lines 373-378.

Reviewer #2 (Remarks to the Author):

The authors have responded well to the reviewers comments, which improved the manuscript substantially. I have no further issues to raise.

We thank the reviewer for their kind comment.

Reviewer #3 (Remarks to the Author):

The authors nicely have addressed all my comments.

We thank the reviewer for their kind comment.

Reviewer #4 (Remarks to the Author):
